# Marine cloud brightening – as effective without clouds

Lars Ahlm[1,2,3*], Andy Jones[4], Camilla W. Stjern[3,5], Helene Muri[3], Ben Kravitz[6], and Jón Egill Kristjánsson[3]

[1]Department of Meteorology, Stockholm University, Stockholm, Sweden.
[2]Bolin Centre for Climate Research, Stockholm University, Stockholm, Sweden.
[3]Department of Geosciences, University of Oslo, Oslo, Norway.
[4]Met Office Hadley Centre, Exeter, UK.
[5]Center for International Climate and Environmental Research—Oslo (CICERO), Oslo, Norway.
[6]Atmospheric Sciences and Global Change Division, Pacific Northwest National Laboratory, Richland, WA, USA.

*Correspondence to*: Lars Ahlm (lars.ahlm@misu.su.se)

**Abstract.** Marine cloud brightening through sea spray injection has been proposed as a climate engineering method for avoiding the most severe consequences of global warming. A limitation of most of the previous modelling studies on marine cloud brightening is that they have either considered individual models, or only investigated the effects of a specific increase in the number of cloud droplets. Here we present results from coordinated simulations with three Earth system models (ESMs) participating in the Geoengineering Model Intercomparison Project (GeoMIP) G4sea-salt experiment. Injection rates of accumulation mode sea spray aerosol particles over ocean between 30°N and 30°S are set in each model to generate a global-mean effective radiative forcing (ERF) of -2.0 W m$^{-2}$ at the top of atmosphere. We find that the injection increases the cloud droplet number concentration in lower layers, reduces the cloud-top effective droplet radius, and increases the cloud optical depth over the injection area. We also find, however, that the global-mean clear-sky ERF by the injected particles is as large as the corresponding total ERF in all three ESMs, indicating a large potential of the aerosol direct effect in regions of low cloudiness. The largest enhancement in ERF due to the presence of clouds occur as expected in the subtropical stratocumulus regions off the west coasts of the American and African continents. However, outside these regions, the ERF is in general equally large in cloudy and clear-sky conditions. These findings suggest a more important role of the aerosol direct effect in sea spray climate engineering than previously thought.

## 1 Introduction

Attempts to lower global emissions of $CO_2$ have so far been mostly unsuccessful. As a result, climate engineering is increasingly being discussed as a way to dampen the climate effects of anthropogenic greenhouse gas emissions. One of the climate engineering methods proposed to counteract global warming is by seeding marine clouds with sea spray aerosol to enhance the number of activated cloud droplets (Latham, 1990). It has been suggested that this could be generated in practice

through the use of unmanned wind-driven vessels spraying sea water into the air (Salter et al., 2008), and as the sea water evaporates it would leave behind sea spray aerosol particles which may be transported into the cloud layer. If the cloud liquid water content in the seeded clouds remains constant, an increase in the cloud droplet number concentration (*CDNC*) will lead to a reduction in cloud droplet size and thereby an increase in droplet surface area and cloud albedo (Twomey, 1977). Increasing

the cloud albedo through this indirect effect of the injected particles is the original idea of sea spray climate engineering, and this method is therefore often referred to as marine cloud brightening. The reduction in cloud droplet size following from an enhanced number of droplets may also lead to a second indirect effect in which the decreased size of the cloud droplets may reduce precipitation and thereby increase the cloud lifetime (Albrecht, 1989).

Earlier modelling studies on sea spray climate engineering investigated the radiative effects of marine cloud brightening

mainly by prescribing an increase in *CDNC* (Latham et al., 2008; Jones et al., 2009; Rasch et al., 2009). However, more recent studies have included the sea salt injection process and the activation of the injected particles to cloud droplets, thereby taking into account radiative effects of both activated cloud droplets and non-activated particles (Jones and Haywood, 2012; Partanen et al., 2012; Alterskjær et al., 2013). As a result, sea spray climate engineering is now sometimes referred to as marine sky brightening (Muri et al., 2015), as it may include radiative impacts of injected particles both through cloud brightening (the

aerosol indirect effect) and due to increased scattering of solar radiation outside clouds (the aerosol direct effect). One of the more recent modelling studies on sea spray climate engineering applied emission patterns to maximize either the direct or the indirect radiative effect of the injected particles, limiting the emission area in both cases to 10% of the ocean (Jones and Haywood, 2012). In that study, maximizing the indirect effect generated the largest radiative impact and resulted in the largest cooling, but it should be noted that the direct effect was of comparable magnitude as that of the indirect effect within the region

specified to maximize the aerosol indirect effect. In another recent modelling study, the aerosol direct effect was estimated to contribute 29% to the total radiative forcing when sea spray climate engineering was assumed to take place over the global oceans (Partanen et al., 2012). In contrast, one recent study indicated a dominant contribution from the aerosol direct effect to the total radiative forcing (Kravitz et al., 2013).

A weakness of almost all of the previous studies on sea spray climate engineering is that they have only considered

individual models. It is therefore uncertain to what extent the results in many of the previous studies are robust, considering the differences in parameterizations across models of e.g. clouds and their interaction with aerosols. Furthermore, results from individual model studies in the past are generally not directly comparable because of discrepancies in the model set-up or in the details of what was actually simulated. Therefore, the idea behind the Geoengineering Model Intercomparison Project (GeoMIP) (Kravitz et al., 2011, 2013) is that model experiments should be standardized, and that an ensemble of multiple

Earth System Models (ESMs) should be executed for a number of climate engineering experiment. By the use of such ensembles, it is possible to estimate an uncertainty in the predicted climate response.

In this study we use three fully coupled atmosphere-ocean ESMs and run the GeoMIP G4sea-salt experiment (see Kravitz et al., 2013; and Sect. 2) focusing on the response of Earth's radiation balance to injection of sea salt particles, both in clear-sky conditions and from changes in cloud properties.

## 2 Methods

### 2.1 Models

Coupled state-of-the-art Earth system models provide the best tools for assessing the climate response to solar climate
engineering. Three fully coupled ESMs, NorESM1-M (Bentsen et al., 2013), GISS-E2-R (Schmidt et al., 2014), and
HadGEM2-ES (Collins et al., 2011), were used in this study. For the atmospheric component, NorESM1-M runs at $1.9° \times 2.5°$
in the horizontal with 26 vertical layers, GISS-E2-R runs at $2° \times 2.5°$ in the horizontal with 20 vertical layers, and HadGEM2-
ES runs at $1.3° \times 1.9°$ in the horizontal with 38 vertical layers. For the ocean component, NorESM1-M runs at $\sim 1° \times 1°$ in the
horizontal with 70 layers, GISS-E2-R runs at $1° \times 1.3°$ in the horizontal with 32 layers, and HadGEM2-ES runs at $1° \times 1°$ in
the horizontal between the poles and 30° latitude with the meridional resolution increasing smoothly to 1/3° at the equator and
with 40 vertical layers.

The treatment of the natural emissions of sea salt is prognostic in NorESM1-M and GISS-E2-R, with emission fluxes
depending on wind speed and sea surface temperatures in NorESM1-M (Struthers et al., 2011), and on wind speed only in
GISS-E2-R (Monahan et al., 1986). HadGEM2-ES uses a diagnostic treatment of natural sea salt aerosol number concentration
with concentrations depending on wind speed (Jones et al., 2001). Hygroscopic growth of aerosol particles is accounted for in
all three models, and this process affects dry removal rates as well as aerosol-radiation interactions. In NorESM1-M,
hygroscopic growth is treated as described by Seland et al. (2008), by applying the form of Köhler equation given in Kirkevåg
and Iversen (2002). In GISS-E2-R, uptake of water by hygroscopic species such as sea salt and sulphate is parameterized in
terms in terms of an external mixture of the dry aerosol and a pure water aerosol with sizes set to reproduce the extinction
efficiency and asymmetry parameters of the solute aerosol at the laboratory wavelength of 633 nm (Schmidt et al., 2006). In
HadGEM2-ES, hygroscopic growth of sea salt and sulphate is modelled following Fitzgerald (1975). NorESM1-M and GISS-
E2-R have fully prognostic treatment of *CDNC*. In HadGEM2-ES, the *CDNC* is a function of sulphate, sea salt and
carbonaceous particle number concentrations (Jones and Haywood, 2012).

Dry deposition of aerosol particles in all three models is parameterized using resistance schemes analogous to electrical
resistance (e.g. Seinfeld and Pandis, 1998). The dry deposition velocity thus depends on particle size. Gravitational settling is
included in the calculation of the dry deposition velocity. Rainout is in all models determined by autoconversion, and include
re-evaporation of precipitation. Wet deposition in NorESM1-M is parameterized as in Iversen and Seland (2002), with an in-
cloud scavenging coefficient defined as the mass fraction of the aerosol mode within the cloud droplet. Wet deposition in
GISS-E2-R and HadGEM2-ES are described in more detail by Koch et al. (2007) and Bellouin et al. (2011), respectively.

### 2.2 Experiments

The following experiments are analysed in this study:

1. RCP4.5: Representative Concentration Pathway 4.5 (Meinshausen et al., 2011), where the total radiative forcing reaches 4.5 W m$^{-2}$ in year 2100, following the CMIP5 protocol (Taylor et al., 2011).

2. G4sea-salt: Follows the experimental design of the Geoengineering Model Intercomparison Project (GeoMIP) G4sea-salt experiment (Kravitz et al., 2013). Sea spray climate engineering is implemented on top of an RCP4.5 scenario to generate a top of atmosphere (TOA) global-mean effective radiative forcing (ERF) of -2.0 W m$^{-2}$. Although sea spray aerosol consists of both sea salt and ocean-derived organic species (e.g. de Leeuw et al., 2011), here we only consider the injection of sea salt particles. The injection is applied at a constant rate in the marine boundary layer between 30°N and 30°S, as this is the area where the largest radiative effects have been predicted from sea salt seeding (Alterskjær et al., 2012; Jones and Haywood, 2012; Kravitz et al., 2013). The sea salt is injected in the lowest model layer of the ESMs, and the injection flux is equally large for each grid cell over the ocean within this latitudinal band. Sea spray climate engineering starts in year 2020 and continues until year 2070, whereupon the simulations are carried on for another 20 years such that the termination effect can be assessed.

3. Fixed sea surface temperature (SST) experiments: The G4sea-salt and RCP4.5 experiments were simulated also with fixed SST, as taken from year 2020 of the RCP4.5 simulation (Kravitz et al., 2013). All other forcing was kept the same as in year 2020 of RCP4.5, with the only difference being increased sea salt emissions. The experiments were run for ten years for each model in order to determine the injection rate of sea salt aerosol in each model required to generate a global-mean ERF of -2.0 W m$^{-2}$ compared to the RCP4.5 scenario. The ERF by the injected particles in these simulations is equal to the change in net total radiation (shortwave + longwave) at the TOA between the G4sea-salt simulation (with sea salt injection) and the RCP4.5 simulation (without sea salt injection). The injection rates required to generate the -2.0 W m$^{-2}$ ERF at the TOA were then applied in the fully coupled simulations between years 2020 and 2070.

The injected sea salt particles within the G4sea-salt experiment have a median dry radius of 0.13 µm in NorESM1-M, 0.44 µm in GISS-E2-R, and 0.10 µm in HadGEM2-ES, equal to the median dry radius of the naturally emitted accumulation mode sea spray particles in each model. The geometric standard deviations of the size distributions are 1.5, 2.0, and 1.9 for NorESM1-M, GISS-E2-R, and HadGEM2-ES, respectively. Size distributions of the injected sea salt particles are shown in Figure 1 for particle number (Fig. 1a), particle surface area (Fig. 1b), and particle mass (Fig. 1c). These size distributions represent the total injection per second within the injection area.

There is large uncertainty in which particle size would be optimal for sea spray climate engineering. The mass scattering efficiency of NaCl particles with a refractive index of 1.544 at a wavelength of 550 nm has its maximum for a particle radius of ~0.3 µm (Seinfeld and Pandis, 1998). However, within the atmosphere hygroscopic growth and condensation of other species like e.g. sulphuric acid will modify the size of the injected particles, which will influence the aerosol direct effect. Latham et al. (2008) estimated that the optimal sea spray dry radius for cloud seeding is in the range of 0.10 to 0.50 µm. In

contrast, Connolly et al. (2014) found using a parcel model that injection of Aitken mode particles would be most efficient, as hygroscopic growth of such injected sea salt particles was shown to significantly enhance the albedo of the cloud layer. Injection of Aitken mode particles, however, generated a positive forcing in NorESM1-M in a previous study by Alterskjær and Kristjánsson (2013), caused by a strong competition effect combined with high critical supersaturation of Aitken mode

particles. Representing sea spray climate engineering in our simulations obviously requires injections that produce a negative forcing. The size of the injected particles in this study is in the same size range as most previous ESM studies on sea spray climate engineering that simulate the aerosol injection (e.g. Alterskjær et al., 2012, 2013; Jones and Haywood, 2012; Korhonen et al., 2010, Muri et al., 2015; and Wang et al., 2011). It should also be mentioned that extensive measurements show that organics contribute substantially to the composition of sea spray aerosol, and in many areas is even the dominant constituent

(e.g. de Leeuw et al., 2011). As sea spray climate engineering would likely produce particles with a similar composition as natural sea spray, the injected particles would thus need to be larger to activate to cloud droplets compared to when assuming pure sea salt as in the study by Connolly et al. (2014). In particular, the presence of organics suppresses hygroscopic growth compared to pure sea salt, which may be relevant since Connolly et al. (2014) found that interstitial particles play an important role in controlling the albedo in their study.

The fully coupled RCP4.5 simulations include two realizations with NorESM1-M, three realizations with GISS-E2-R, and four realizations with HadGEM2-ES. The fully coupled G4sea-salt simulations include two realizations with NorESM1-M, three realizations with GISS-E2-R, and one realization with HadGEM2-ES.

**3 Results and discussion**

A key variable in the models when considering sea spray climate engineering, is the amount of low clouds over the ocean, in particular subtropical stratocumulus clouds off the west coasts of North America, South America, and southern and northern Africa. These regions have been assessed to be most susceptible to brightening (Salter et al., 2008; Alterskjær et al., 2012; Jones and Haywood, 2012). Figure 2 shows the low-level cloud fraction below 850 hPa for NorESM1-M (Fig. 2a) and

HadGEM2-ES (Fig. 2c), and below 600 hPa for GISS-E2-R (Fig. 2b), averaged over years 2020-2030 in the RCP4.5 scenario. Here we use the assumption of random overlapping cloud layers for the estimates of the cloud cover. NorESM1-M (Fig. 2a) and HadGEM2-ES (Fig. 2c) capture the maxima in low-level cloud cover associated with the subtropical high pressure cells in the eastern parts of the Pacific Ocean and the Atlantic Ocean (e.g. Rossow and Schiffer, 1999). The reason for including layers higher than 850 hPa in the estimate of low-level cloud cover for GISS-E2-R is that for the region west of Peru the model

reaches its maximum in cloud cover slightly above 850 hPa. From Fig. 2 it is clear that the low-cloud amounts over tropical and subtropical ocean are considerably lower in GISS-E2-R than in NorESM1-M and HadGEM2-ES, in particular when it comes to stratocumulus clouds in the subtropical high pressure cells in the eastern parts of the Pacific Ocean and Atlantic Ocean. This needs to be taken into account in the assessment of the impact of sea spray climate engineering in GISS-E2-R.

## 3.1 Effective radiative forcing by the injected particles

The sea salt injection rates between 30°N and 30°S required to generate a global-mean ERF of -2.0 W m$^{-2}$ at the TOA are 250 Tg yr$^{-1}$ in NorESM1-M, 590 Tg yr$^{-1}$ in GISS-E2-R, and 200 Tg yr$^{-1}$ in HadGEM2-ES. The fact that GISS-E2-R requires a larger injection rate than the two other ESMs is likely due to the larger dry radius of the injected particles in GISS-E2-R (0.44 µm) than in NorESM1-M (0.13 µm) and HadGEM2-ES (0.10 µm). This means that a specific injection rate in GISS-E2-R results in fewer particles than in the two other ESMs (Fig. 1a). The smaller amount of low clouds in GISS-E2-R (Fig. 2b) may also be a contributing factor to the larger injection rates required in this model. The injection rates in this study are close to the rate reported by Partanen et al. (2012) who obtained a -5.1 W m$^{-2}$ global-mean ERF in the aerosol-climate model ECHAM5.5-HAM2 from wind speed-dependent global sea salt injections at a rate of 440 Tg yr$^{-1}$. Our injection rates are also similar to those reported by Alterskjær et al. (2013) who applied gradually increasing sea salt injection rates between 30°N and 30°S in three different ESMs to keep the TOA radiative forcing of an RCP4.5 scenario at the 2020 level for 50 years. The radiative forcing change within the RCP4.5 scenario between 2020 and 2070 is +1.64 W m$^{-2}$. During the last decade of their simulations, the injection rates required varied between 266 and 560 Tg yr$^{-1}$ across their three models.

The global-mean ERF by the injected sea salt particles, for the rates given above, is relatively constant at -2.0 W m$^{-2}$ throughout the 10 years fixed SST simulation in all three ESMs (Fig. 3a). The radiative fluxes in the ESMs are calculated also for clear-sky conditions. These clear-sky radiative fluxes can be used to determine the clear-sky global-mean ERF (Fig. 3b). This variable is not equal to the aerosol direct effect of the injected particles, because the aerosol direct effect is larger in clear-sky conditions than when clouds are present. This is because most of the injected particles are located below cloud base when clouds are present, which reduces the aerosol direct effect due to the high albedo of most clouds. Nevertheless, it is interesting to note that the clear-sky global-mean ERF (Fig. 3b) is almost equal to the total global-mean ERF (Fig. 3a) throughout the 10 years in the three ESMs, indicating a large potential of the aerosol direct effect in regions of low cloudiness. Although we cannot estimate the contribution of the aerosol direct effect to the total ERF from Fig. 3, it is evident that sea spray climate engineering can be effective even without clouds.

The clear-sky ERF by the injected particles in Fig. 3b is of comparable magnitude for the three models, despite the higher sea salt mass injection rates and larger size of the injected particles in GISS-E2-R compared to the other two models. The surface area size distribution (Fig. 1b) is closely related to the amount of light scattered by the sea salt particles and thereby the clear-sky ERF in Fig. 3b. For a full description of Mie scattering, however, one needs to take into account also variations in the scattering coefficient with particle size, which is done in the radiative transfer calculations in the models. The total particle number injections (integrated over the particle number size distributions in Fig. 1a) are $1.8\times10^{20}$, $2.7\times10^{18}$, and $1.1\times10^{20}$ s$^{-1}$ for NorESM1-M, GISS-E2-R, and HadGEM2-ES, thus almost two orders of magnitude smaller number injection in GISS-E2-R compared to the other models. The corresponding particle surface injections (integrated over the particle surface distributions in Fig. 1b) are $5.2\times10^{7}$, $1.7\times10^{7}$, and $3.1\times10^{7}$ m$^2$ s$^{-1}$ for NorESM1-M, GISS-E2-R, and HadGEM2-ES. Thus, although the difference in total particle number injection between GISS-E2-R and the other two models is large, the difference

in total particle surface area injection is considerably smaller. Hygroscopic growth is accounted for in all the three models (Sect. 2.1) and this process will modify the injected particle size distributions in the atmosphere, since the relative humidity within the injection area is generally above the deliquescence relative humidity for sea salt. The light scattering enhancement factor (e.g. Titos et al., 2016) describes the relative increase in aerosol light scattering at a certain relative humidity compared to dry conditions. This parameter is not diagnosed in the models for the injected sea salt particles, but decreases with increasing particle dry diameter for a certain relative humidity (e.g. Zieger et al., 2013). This means that hygroscopic growth of the injected particles is expected to generate a larger increase in clear-sky ERF in NorESM1-M and HadGEM2-ES than in GISS-E2-R, since the injected particles are larger in GISS-E2-R than in the two other models. The main reason that sea salt injections in GISS-E2-R still generates a clear-sky ERF as large as the other two models, or even slightly larger (Fig. 3b), is likely due to GISS-E2-R having the lowest background clear-sky atmospheric optical depth of the three models (not shown). This means that GISS-E2-R is more sensitive to injections than the two other models.

The effective radiative forcing by the injected particles at the TOA varies spatially between -2.0 and -10 W m$^{-2}$ across the injection area in the three ESMs (Fig. 4). The mean values over the injection area are -4.3 W m$^{-2}$ in NorESM1-M, -4.9 W m$^{-2}$ in GISS-E2-R, and -4.7 W m$^{-2}$ in HadGEM2-ES. The injection area here, and for later calculations, represents all grid cells over ocean between 30°N and 30°S. In NorESM1-M (Fig. 4a), maximum ERF appear in the stratocumulus regions off the west coasts of northern South America and southern Africa (Fig. 2a), locally exceeding -10 W m$^{-2}$. This means that the ERF over these regions is a factor of 2-3 larger than the average over the injection area. The location of these maxima is in agreement with the studies by Jones and Haywood (2012) and Partanen et al. (2012) who observed a strong aerosol indirect effect in these areas from sea spray climate engineering. The ERF maximum off the west coast of southern Africa is pronounced also in HadGEM2-ES (Fig. 4c), although weaker in forcing than in NorESM1-M. In addition, there are maxima in ERF in the marine stratocumulus regions west of northern Africa and west of Australia for both NorESM1-M and HadGEM2-ES. Jones and Haywood (2012) saw a strong aerosol indirect effect from sea spray climate engineering in these regions in the HadGEM2-ES model.

Although the effective radiative forcing by the injected particles in NorESM1-M and HadGEM2-ES is at maximum over some of the marine subtropical stratocumulus regions previously identified as optimal for marine cloud brightening, the ERFs in Fig. 4 are not as dominated by these regions as in the study by Partanen et al. (2012) with ECHAM5.5-HAM2. In that study, the maximum ERF of their sea spray climate engineering exceeded -30 W m$^{-2}$ in the stratocumulus regions west of Peru and southern Africa, whereas the mean ERF outside these regions was around -5 W m$^{-2}$. This means that the ERF by the injected particles in the subtropical stratocumulus regions was more than a factor of six higher than the typical ERF outside these regions. Such a large difference in ERF between subtropical stratocumulus regions and other regions within the injection area is not seen here. For NorESM1-M and HadGEM2-ES the sea salt injection also generates a strong ERF over large regions of the central and western parts of the Pacific Ocean where a low cloud-weighted susceptibility to sea salt injections (Alterskjær et al., 2012), and a strong aerosol direct effect from sea spray climate engineering (Jones and Haywood, 2012), have been identified. The correlation between the strength of the effective radiative forcing and low-level cloud cover (as defined in Fig.

2), when including all grid cells over ocean within the injection area, is weak for these two models (the Pearson correlation coefficient $r$ is equal to 0.28 and 0.16 for NorESM1-M, and HadGEM2-ES respectively). Thus, over the injection area as a whole, the presence of low-level clouds gives no clear advantage for obtaining a large ERF from sea spray climate engineering.

Although GISS-E2-R (Fig. 4b) has maxima in ERF in the same subtropical stratocumulus regions as the other two models, there is less horizontal variability in ERF in GISS-E2-R. An exception is the Intertropical Convergence Zone (ITCZ) where the ERF is considerably weaker, likely due to the large amounts of high clouds in these regions (not shown). The presence of middle to high-level clouds is not optimal for sea spray climate engineering as these clouds block out some of the incoming solar radiation, and make a negligible contribution to the aerosol indirect effect. A weaker ERF along the ITCZ can be seen to some extent also over the Pacific in NorESM1-M (Fig. 4a). The more homogeneous ERF field for GISS-E2-R compared to the two other models is likely due to the smaller amount of low-level clouds in GISS-E2-R compared to the two other ESMs (Fig. 2). This means that the aerosol direct effect likely contributes more to the total ERF in GISS-E2-R leading to less horizontal variations in ERF. This hypothesis of a low contribution of the aerosol indirect effect to the ERF in GISS-E2-R is supported by the absence of correlation between the strength of the ERF and low-level cloud cover ($r = -0.10$) for this model.

Figure 5 shows the ratio of the total ERF to clear-sky ERF at the TOA for each of the three models. This figure provides information on whether the clouds that are present increase the ERF by the injected particles compared to clear-sky conditions. Red-coloured areas indicate an increased ERF when clouds are present, and thereby an effective aerosol indirect effect, whereas blue-coloured regions indicate an enhanced ERF for clear-sky conditions. The impact of the subtropical stratocumulus clouds on the ERF by the injected particles, relative to clear-sky conditions, is largest in HadGEM2-ES (Fig. 5c), with the ratio of total ERF to clear-sky ERF locally being higher than 4:1 in regions west of California and Mexico, west of southern Africa, and west of Australia. In NorESM1-M (Fig. 5a), the corresponding enhancement in ERF in these regions due to the presence of low clouds is considerably smaller, although with a ratio locally above 3:1 in the Atlantic region west of northern Africa. In GISS-E2-R (Fig. 5b), the maximum values of the total ERF to clear-sky ERF ratio appear in the same subtropical high pressure regions as in the other models, although much less pronounced due to the smaller amount of low-level clouds in this model. For GISS-E2-R, there are regions along the ITCZ where the presence of clouds reduces the ERF by the injected particles (blue-coloured regions), likely due to the high presence of high-level clouds in these regions, as discussed above. Total ERF to clear-sky ERF ratios lower than one along the ITCZ indicate that the aerosol direct effect of the injected particles in clear-sky conditions is larger than the total radiative effect of the injected particles when clouds are present, and such ratios appear locally also in the other two ESMs.

In summary, the presence of low clouds in the subtropical high pressure regions have the effect of increasing the ERF by the injected particles compared to clear-sky conditions, and this enhancement in ERF due to the aerosol indirect effect is most pronounced in HadGEM2-ES. However, in most other regions within the area of sea salt injection, the ratio of total ERF to clear-sky ERF is close to one in all the models, which indicates that the presence of clouds in most regions does not significantly increase the ERF compared to clear-sky conditions. This finding, together with the relatively small horizontal variability in ERF compared to Partanen et al. (2012) and weak or non-existent correlations between ERF and low-level cloud cover, suggest

that the aerosol direct effect probably makes a larger contribution to the total ERF in this study compared to the study by Partanen et al. (2012) where the aerosol direct effect contributed 29% to the total ERF by sea spray climate engineering.

## 3.2 Coupled simulations

### 3.2.1 Change in sea salt concentrations, cloud properties, and atmospheric circulation

The injection rates generating a global-mean effective radiative forcing of -2.0 W m$^{-2}$ at the TOA in the simulations with fixed SST were applied in the fully coupled G4sea-salt simulations between 2020 and 2070 in the three ESMs. These sea salt

injections elevate the sea salt mass concentration within the injection area in all the models compared to the RCP4.5 scenario (Fig. 6). As mentioned in Sect. 3.1, the injection rate in GISS-E2-R was 2-3 times higher than in the two other ESMs, which explains the larger enhancement in mass concentration in GISS-E2-R compared to the other models. Despite equal sea salt flux increase in all grid cells within the injection area, there are large spatial variations in the increase in sea salt concentration in the lowest model layer in all the models. This is due to differences in precipitation, boundary layer depth, and horizontal

and vertical transport across different regions. In NorESM1-M (Fig. 6a), comparatively large increases in sea salt concentration occur in the subtropical high pressure regions. This is likely a combined effect of low precipitation, thin boundary layer, and generally little vertical mixing in these regions compared to regions with more convection. Similar patterns can be seen in GISS-E2-R (Fig. 6b) and in part in HadGEM2-ES (Fig. 6c). HadGEM2-ES has further peak increases in sea salt concentrations closer to the equator, which could indicate either more efficient aerosol transport equatorward by trade winds, or less efficient

wet removal in the ITCZ.

One of the advantages of simulating sea spray climate engineering in ESMs through sea salt aerosol emissions, compared to just increasing the *CDNC*, is that the cloud droplet activation process is taken into account. Previous studies have shown that injection of sea spray particles in some circumstances may actually reduce the *CDNC* due to increased competition for water vapour and reduced activation of background aerosol particles (Korhonen et al., 2010; Alterskjær et al., 2012). Alterskjær

and Kristjánsson (2013) showed in a single-model study that while the injection of accumulation mode particles increased the *CDNC*, the injections of Aitken or coarse mode particles could have the opposite effect with a reduction in *CDNC*. As mentioned in Sect. 2.2, the injected particles in this study are accumulation mode particles with a median dry radius between 0.10 and 0.44 µm. The background *CDNC* within the injection area at an altitude of ~1000 m averaged over 2035-2065 for RCP4.5 varies for NorESM1-M from 10-20 cm$^{-3}$ in the remote areas of Pacific and reaches a maximum of ~100 cm$^{-3}$ south of

Mexico, west of Northern Africa, south-east of China, and over the northern parts of the Indian Ocean. HadGEM2-ES has its maxima in *CDNC* at similar locations within the injection area. However, HadGEM2-ES has somewhat higher concentrations with a typical *CDNC* of 20-40 cm$^{-3}$ in the remote Pacific Ocean and *CDNC* reaching 250 cm$^{-3}$ at coastal locations closer to continental sources. GISS-E2-R has higher background *CDNC* than the other models, with concentrations of 50-100 cm$^{-3}$ in the remote Pacific Ocean and concentrations higher than 1000 cm$^{-3}$ in some coastal regions influenced by continental sources.

Whereas NorESM1-M and HadGEM2-ES simulate *CDNC* close to estimates using MODIS (Moderate Resolution Imaging Spectroradiometer) data for cloud-top *CDNC* (e.g. Wood, 2012), GISS-E2-R predicts higher background *CDNC* than estimated from MODIS.

As shown in Fig. 7, the sea salt injection enhances the *CDNC* in lower layers within the whole injection area in all three ESMs. The mean percentage increase in *CDNC* within the injection area averaged for the period 2035-2065 (only grid cells over ocean included) is 153% in NorESM1-M, 42% in GISS-E2-R, and 89% in HadGEM2-ES (Table 1). The largest enhancements in *CDNC* generally occur in regions where the background *CDNC* is low. The smaller percentage increase in *CDNC* in GISS-E2-R compared to the other two models is likely due to the higher background *CDNC* in GISS-E2-R. Over the Arctic region, there is a relatively large reduction in *CDNC* in NorESM1-M (Fig. 7a) and HadGEM2-ES (Fig. 7c). However, the *CDNC* in the Arctic region is as low as ~1 cm$^{-3}$, which implies that a very small absolute change in concentration can result in a large relative change in *CDNC*. The mechanism for the reduction of *CDNC* in the Arctic is likely related to the cooling induced by the sea-salt: the cooling increases the sea-ice cover in the Arctic and therefore reduces the source of natural sea salt and Dimethyl sulphide (DMS), both of which cause a reduction in *CDNC*. The cooling also reduces the liquid water in the clouds, which may also contribute to the reduction in *CDNC*, as this variable represents the number concentration of cloud liquid water particles in the air.

As expected, the cloud-top effective droplet radius, $r_e$, is reduced due to the sea salt injection over the whole injection area (Figs. 8a and c). The mean reductions in $r_e$ within the injection area are -8.6% for NorESM1-M and -6.4% for HadGEM2-ES ($r_e$ could not be diagnosed in GISS-E2-R) (Table 1). Although the change in cloud water path (vertically-integrated cloud water content including both liquid water and ice) due to the sea spray climate engineering is more than 15% locally in all three ESMs (shown as multi-model mean in Fig. 9b), the mean changes globally and over the injection area are less than 2% in all three models (Table 1). Interestingly, there is no correlation between the change in *CDNC* and the change in cloud water path within the injection area for mean values of these variables over years 2035-2065. The Pearson correlation coefficient *r* for this relation is 0.09 for NorESM1-M, -0.24 for GISS-E2-R, and -0.09 for HadGEM2-ES. The lack of such a correlation and the fact that the mean change in cloud water path within the injection area is small, and even negative in two of the models (Table 1), indicate that the second aerosol indirect effect is weak (Malavelle et al., 2017). Local changes in cloud water path within the injection area appear instead to be linked to changes in the atmospheric circulation. This is seen in the correlation between the change in cloud water path and the change in omega-vertical velocity (Fig. 9a; *r* = -0.70 for NorESM1-M, *r* = -0.59 for GISS-E2-R, and *r* = -0.63 for HadGEM2-ES). The negative *r*-coefficients here indicate an increasing cloud water path under increasing upward motion in the atmosphere.

The cloud optical depth ($\tau$) can be estimated from the cloud liquid water path (*LWP*) and the cloud droplet effective radius at cloud-top ($r_e$) through the following relation (Stephens, 1978):

$$\tau \approx \frac{3}{2} \frac{LWP}{r_e} \tag{1}$$

*LWP* have the units of g m$^{-2}$ and $r_e$ is in μm. Note that Table 1 gives the change in cloud water path including ice whereas *LWP* in Eq. 1 only refers to liquid water. As the estimate of $\tau$ using Eq. 1 requires the variable $r_e$, $\tau$ could only be estimated for NorESM1-M (Fig. 8b) and HadGEM2-ES (Fig. 8d). As seen in Fig. 8, the sea salt injection results in an increase in $\tau$ in most

regions within the injection area in both ESMs. The mean increase in $\tau$ over the injection area is 10% for NorESM1-M and 6% for HadGEM2-ES (Table 1). However, locally $\tau$ increases by more than 20% in both models. An anti-correlation between the relative change in $\tau$ and the corresponding change in $r_e$ exists, although moderate to weak, in the two models (Figs 10a-b). A negative correlation coefficient is expected due to the Twomey effect. The correlation between the relative changes in $\tau$ and *CDNC* is even weaker (Fig. 10c-d). By far the strongest correlation is the one between the relative changes in $\tau$ and *LWP* (Fig.

10e-f). Thus, despite the increase in *CDNC* due to the sea salt injection, it seems that local changes in $\tau$ are controlled largely by changes in *LWP*, which in turn are caused mainly by changes in the atmospheric circulation.

Figure 9 shows the multi-model mean changes in omega-vertical velocity (a), cloud water path (b), and precipitation (c). In large regions over the Eastern Pacific Ocean, reduced ascent (or increased subsidence) (Fig. 9a) is accompanied by reductions in cloud water path (Fig. 9b) and precipitation (Fig. 9c). Enhanced ascent over e.g. Africa, northern South America,

and in the South Pacific Convergenze Zone, on the other hand, coincide with increased cloud water path and precipitation. These patterns of enhanced cloud water, precipitation, and atmospheric upward motion over low-latitude continents combined with reduced cloud water, precipitation, and ascent over some low-latitude ocean regions have been reported previously by Bala et al. (2011), Alterskjær et al. (2013), Niemeier et al. (2013), Crook et al. (2015), and Stjern et al. (2017). This is a result of reduced absorption of solar radiation over ocean where sea salt concentrations are elevated while continental regions are

left less affected, increasing the land – sea gradient over the tropics. This induces enhanced convection over land and thereby increased cloud formation and precipitation, and reduced cloud formation over ocean due to reduced upward motion or increased subsidence. Furthermore, the increase in upward motion and cloud water content north-east of Australia, and the reduction in these variables over the Eastern Pacific Ocean west of South America, indicate a strengthening of the Pacific Walker cell and South Pacific Convergence Zone.

In summary, the aerosol indirect effect of the injected sea salt particles can be seen in the mean increase in *CDNC*, mean decrease in cloud-top effective droplet radius, and mean increase in cloud optical depth over the injection area. However, in these fully coupled simulations the aerosol direct and indirect effects of the injected sea salt particles also cause changes in the atmospheric circulation that generate a redistribution of cloud water, with increasing cloud water and precipitation in regions of enhanced atmospheric ascent and decreasing cloud water and precipitation in regions of decreased atmospheric upward

motion. Within the injection area, the local response in cloud optical depth is controlled to a larger extent by these changes in cloud water, than by changes in *CDNC* or $r_e$. This means that it is not necessarily the regions that are exposed to the largest aerosol indirect effect of the injected particles that are exposed to the largest enhancement in cloud albedo.

**3.2.2 Change in net SW radiation at the TOA**

The global-mean difference in net SW radiation at the TOA (Fig. 11a) between G4sea-salt and RCP4.5 is rather constant at -2.0 W m$^{-2}$ throughout the 50 years of sea spray climate engineering in NorESM1-M and GISS-E2-R, hence similar in magnitude to the global-mean ERF by the sea salt injection. Thus, in these two ESMs a constant sea salt injection in time increases the planetary albedo with a factor that is roughly constant in time, despite slow feedbacks being included in these fully coupled simulations. In HadGEM2-ES, the difference in net SW radiation at the TOA between G4sea-salt and RCP4.5 is increasing somewhat during the 50 years of sea spray climate engineering, which means that a constant sea salt injection rate in HadGEM2-ES generates a slowly increasing planetary albedo. Positive cloud feedback should be contributing to this in HadGEM2-ES, which will act to increase the radiative effect of climate engineering over time, in contrast to the negative cloud feedback in NorESM1-M (Andrews et al., 2012). However, there is some indication of an increasing difference in net SW radiation between G4sea-salt and RCP4.5 over time in HadGEM2-ES also for the clear-sky fluxes (Fig. 11b), indicating a contribution from the sea ice albedo feedback. The reduction in net SW radiation at the TOA over the Arctic region, caused by the sea spray climate engineering, is larger in HadGEM2-ES (Fig. 12c) than in the two other ESMs (Figs.12a-b), which indicates that the sea ice albedo feedback is strongest in HadGEM2-ES. HadGEM2-ES also has a larger reduction in surface temperature than the other two models for the Arctic region (not shown). However, reductions in global-mean surface temperature are very similar in the three models (Table 1).

Whereas the global mean changes in net SW radiation at the TOA shown in Figs. 11a-b are to some extent influenced by changes in surface albedo, the corresponding changes over the injection area over ocean between 30°N and 30°S are only due to atmospheric changes (Figs. 11c-d). As expected, the reductions in net SW radiation are on average larger between 30°N and 30°S, where the sea salt injection occurs, than globally. The total change in net SW radiation over the injection area (Fig. 11c) is rather constant with time in NorESM1-M and GISS-E2-R, similar to the global mean curves in Fig. 11a, but slowly increasing with time in HadGEM2-ES. The change in clear-sky net SW radiation over the injection area (Fig. 11d) is rather constant with time in all three ESMs. Similar to the ERF in Fig. 3, the change in clear-sky net SW radiation over the injection area is almost equal to the total change in net SW radiation in the three ESMs, again indicating a large potential of the aerosol direct effect in regions of low cloudiness. In GISS-E2-R and NorESM1-M, the change in net SW radiation is even larger in clear-sky conditions than in total.

## 4 Conclusions

In this study, we have analysed the GeoMIP G4sea-salt experiment using three different ESMs: NorESM1-M, GISS-E2-R, and HadGEM2-ES. Sea spray climate engineering is applied on top of the RCP4.5 scenario between years 2020 and 2070, with sea salt injection rates set to generate a global-mean ERF of -2.0 W m$^{-2}$.

Although sea spray climate engineering is often referred to as marine cloud brightening, we find that the global-mean clear-sky ERF is as large as the total ERF in all three ESMs, indicating the large potential of the aerosol direct effect in regions

of low cloudiness. The largest regional enhancement in ERF due to the presence of clouds, compared to the ERF in clear-sky conditions, occur as expected in the subtropical stratocumulus regions off the west coasts of the American and African continents. However, in most regions outside these subtropical regions, the clear-sky ERF is as large as the total ERF. Furthermore, the correlation between low-level cloud cover and the strength of the ERF by the injected particles within the injection area is weak or non-existent in the models. These factors together indicate that with the exception of the subtropical stratocumulus regions, sea spray climate engineering is as efficient in clear-sky conditions as in cloudy-sky conditions.

The aerosol indirect effect of the injected particles is seen in the increase in *CDNC*, reduction in $r_e$, and increase in cloud optical depth over the injection area. However, sea spray climate engineering also causes changes in the atmospheric circulation, which results in a redistribution of cloud water. We find that the local response of the cloud optical depth depends to a larger extent on changes in the *LWP* than on changes in *CDNC* or in $r_e$.

These results show that many important secondary effects on clouds are neglected if sea spray climate engineering is investigated by the simplified method of increasing the number of cloud droplets, as has been done previously in a number of studies (Latham et al., 2008; Jones at al., 2009; Rasch et al., 2009), or when considering injection in a limited area (Jones and Haywood, 2012). The results here may also have implications for which regions may be most effective in generating a cooling from sea spray injection, as the aerosol direct effect likely plays a more important role than previously thought.

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

Ben Kravitz, were supported by the NASA High-End Computing (HEC) Program through the NASA Center for Climate Simulation (NCCS) at Goddard Space Flight Center. The Pacific Northwest National Laboratory is operated for the U.S. Department of Energy by Battelle Memorial Institute under contract DE-AC05-76RL01830. We also thank all participants of the Geoengineering Model Intercomparison Project and their model development teams, CLIVAR/WCRP Working Group on Coupled Modeling for endorsing GeoMIP, and the scientists managing the Earth System Grid data nodes who have assisted with making GeoMIP and CMIP5 output available.

**Table 1:** Mean percentage changes in CDNC, Cloud-Top Effective Radius, Cloud Water Path, Cloud Cover, Cloud Optical Depth, Precipitation, and Surface Air Temperature (°C) due to sea spray climate engineering. The changes represent the percentage difference between G4sea-Salt (with climate engineering) and RCP4.5 (without climate engineering) averaged over the period 2035-2065 for the injection area and globally. The change in CDNC represents the change in cloud droplet number concentration within the model layer below 700 hPa with maximum concentration. Cloud-Top Effective Radius and Cloud Optical Depth could only be diagnosed for NorESM1-M and HadGEM2-ES. Cloud Optical Depth has been estimated using Eq. 1 (Stephens, 1978).

| | NorESM1-M | | GISS-E2-R | | HadGEM2-ES | |
|---|---|---|---|---|---|---|
| | Injection area mean | Global mean | Injection area mean | Global mean | Injection area mean | Global mean |
| CDNC (%) | 153 | 65 | 28 | 15 | 89 | 36 |
| Cloud-Top Effective Radius (%) | -8.6 | -5.5 | - | - | -6.4 | -3.8 |
| Cloud Water Path (%) | 0.53 | -0.11 | -1.3 | -1.9 | -1.3 | -1.4 |
| Cloud Cover (%) | -2.8 | -2.0 | 0.05 | 0.11 | -0.11 | -0.24 |
| Cloud Optical Depth (%) | 10 | 5.9 | - | - | 5.7 | 2.1 |
| Precipitation (%) | -3.7 | -2.7 | -1.2 | -1.1 | -2.6 | -2.0 |
| Surface Temperature (°C) | -0.68 | -0.54 | -0.83 | -0.62 | -0.76 | -0.62 |

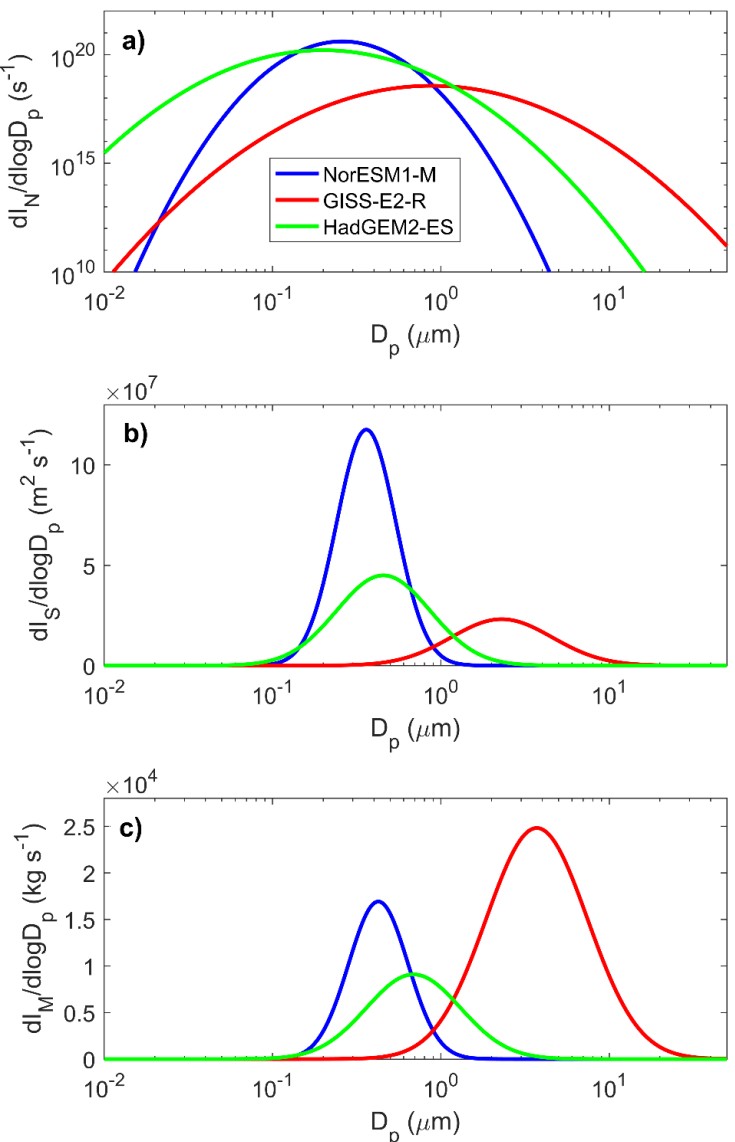

**Figure 1.** Size distributions for total sea salt injections (30°N and 30°S) of a) particle number $I_N$, b) particle surface area $I_S$, and c) particle mass $I_M$ for NorESM1-M (blue), GISS-E2-R (red), and HadGEM2-ES (green).

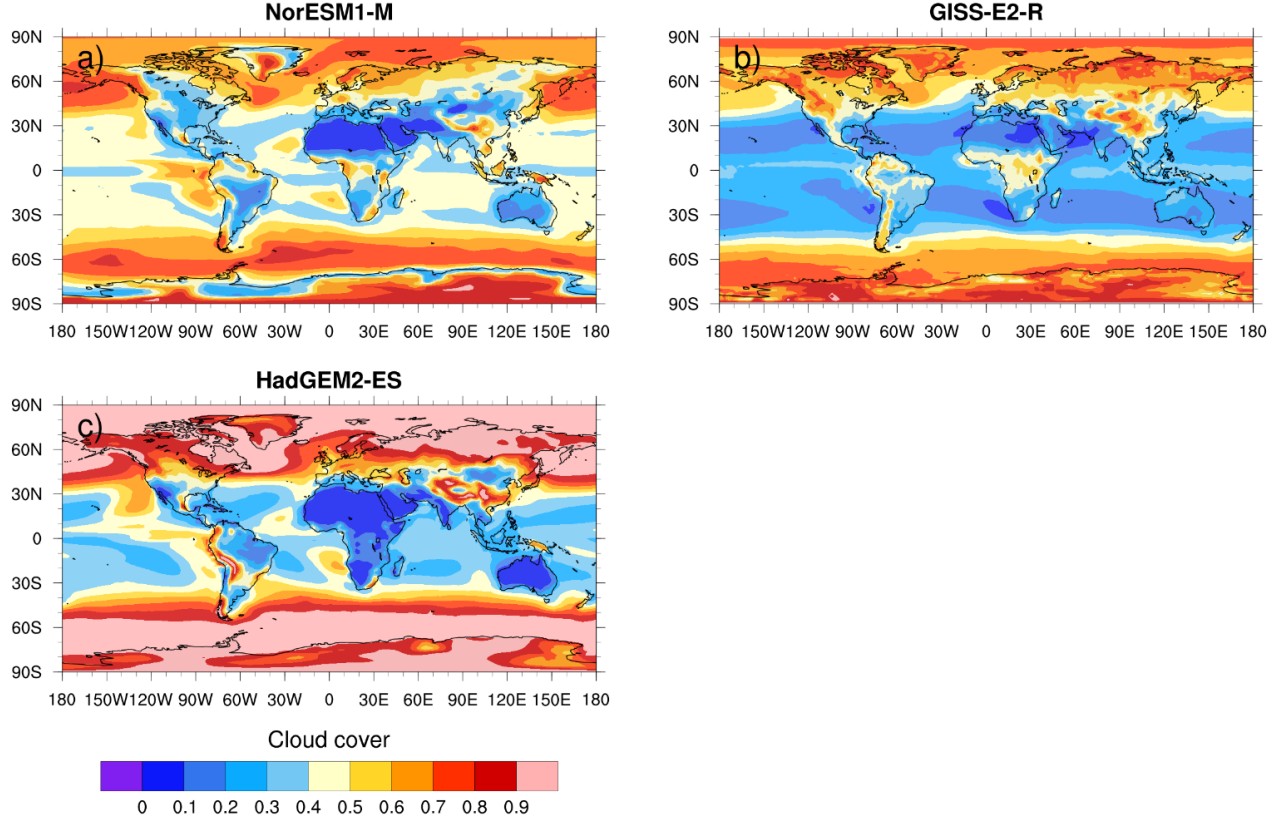

**Figure 2.** Cloud fraction for low clouds averaged over 2020-2030 within the RCP4.5 scenario for **a)** NorESM1-M, **b)** GISS-E2-R, and **c)** HadGEM2-ES. Cloud fractions have been estimated by assuming random overlapping for layers below 850 hPa (a and c) and below 600 hPa (b).

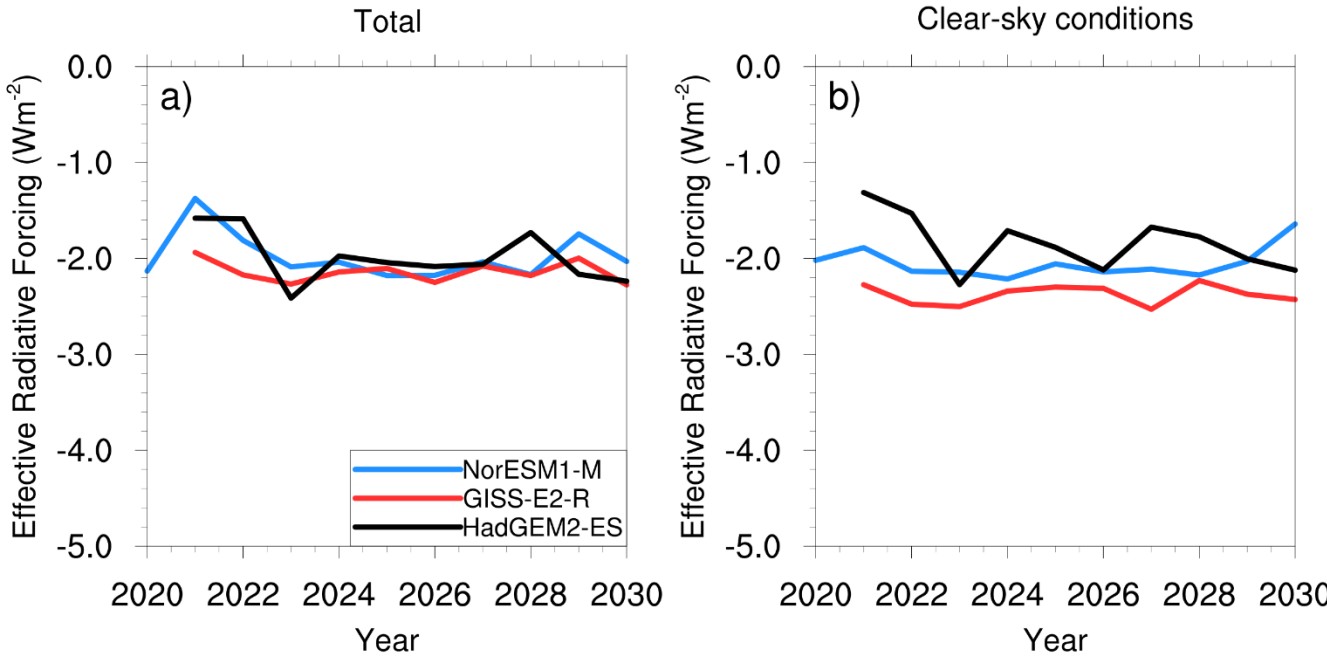

**Figure 3.** Global-mean TOA effective radiative forcing of the injected particles in total **(a)** and in clear-sky conditions **(b)**. The ERF for each model was determined from 10-year simulations with fixed SST with and without sea salt injection.

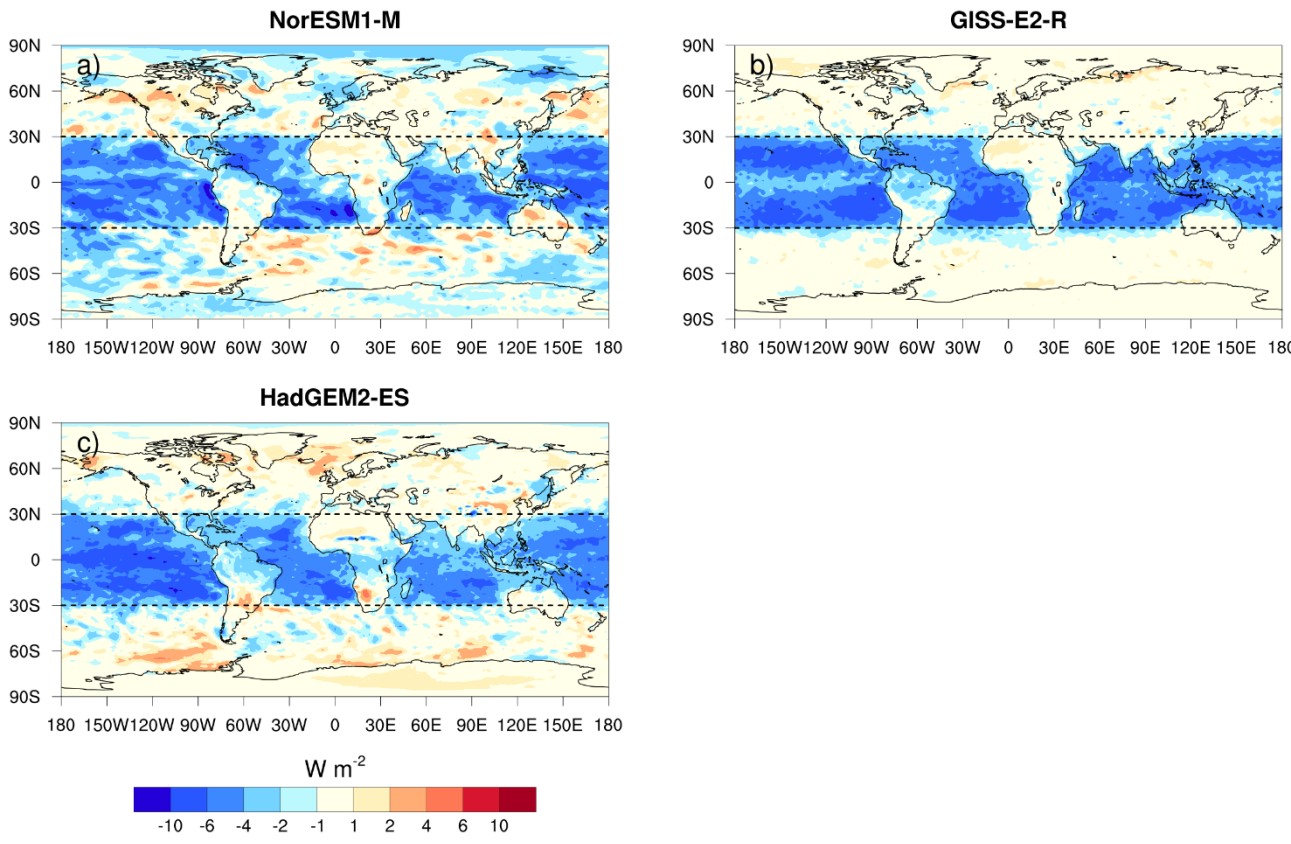

**Figure 4.** TOA mean effective radiative forcing over the 10 years of simulation with fixed SST for **a)** NorESM1-M, **b)** GISS-E2-R, and **c)** HadGEM2-ES.

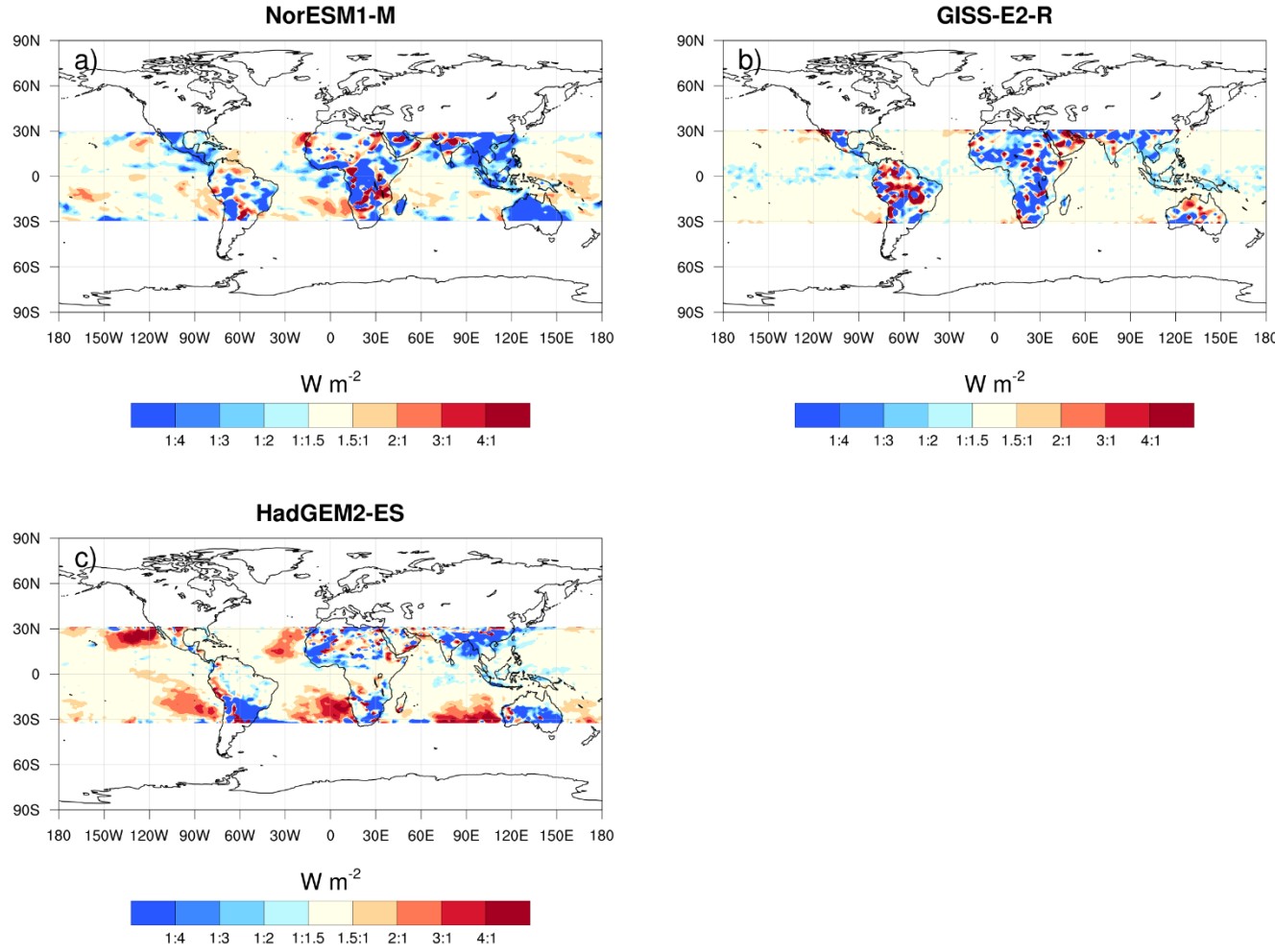

**Figure 5.** The ratio of the total ERF to the clear-sky ERF at the TOA averaged over the 10 years of simulation with fixed SST for **a)** NorESM1-M, **b)** GISS-E2-R, and **c)** HadGEM2-ES.

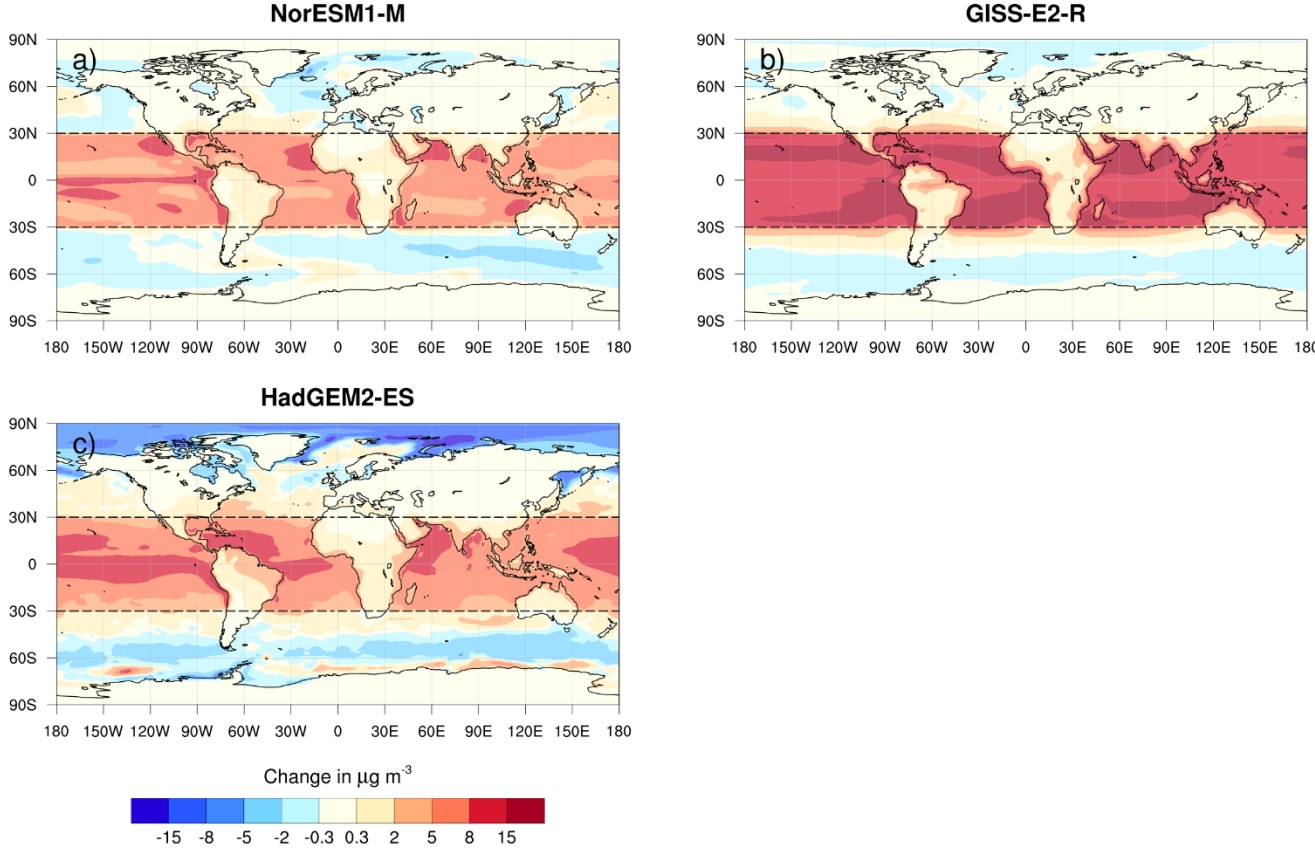

**Figure 6.** Mean difference in sea salt mass concentration in the lowest model layer between G4sea-Salt and RCP4.5 averaged over 2035 and 2065, for **a)** NorESM1-M, **b)** GISS-E2-R, and **c)** HadGEM2-ES.

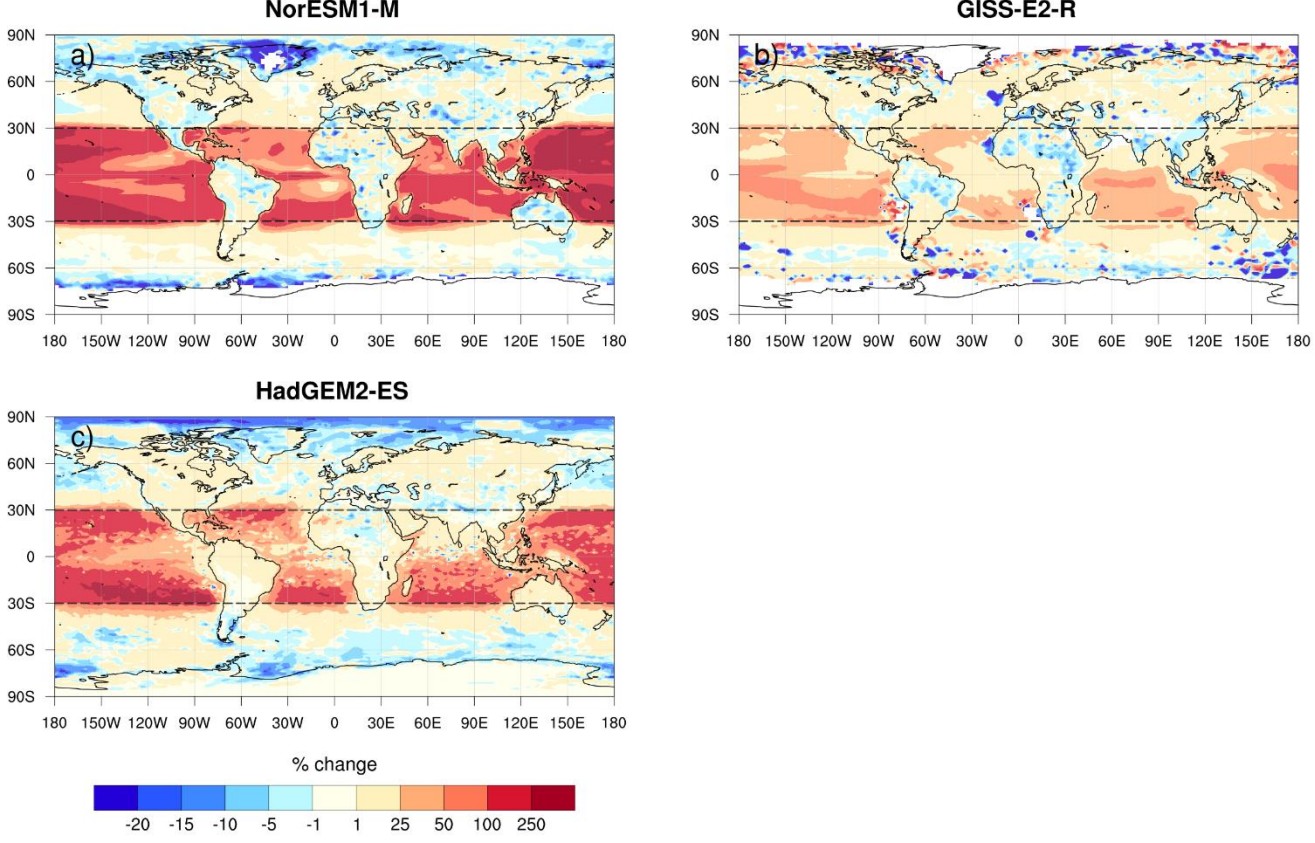

**Figure 7.** Mean relative change in CDNC due to sea salt injection for NorESM1-M (**a**), GISS-E2-R (**b**), and HadGEM2-ES (**c**). CDNC represents the cloud droplet number concentration within the model layer below 700 hPa with maximum concentration, and the maps represent averages over 2035-2065.

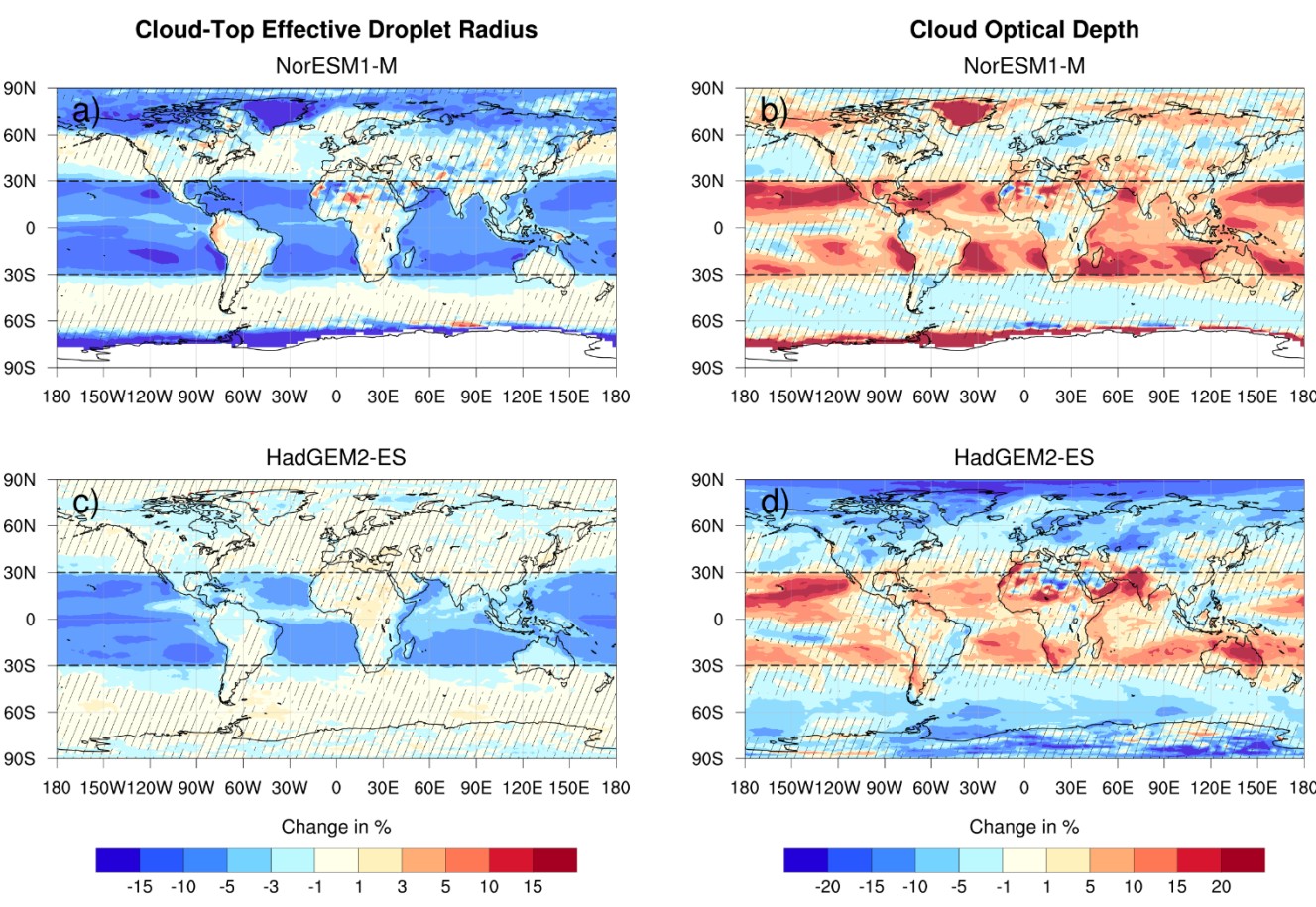

**Figure 8.** Mean relative change between G4sea-Salt and RCP4.5 in Cloud-Top Effective Radius for NorESM1-M **(a)** and HadGEM2-ES **(c)**, and in Cloud Optical Depth for NorESM1-M **(b)** and HadGEM2-ES **(d)**. The maps represent average change due to sea spray climate engineering over the period 2035-2065. Hatching denotes areas where changes are not significant at the 95% confidence level (Student t-test with respect to variance of annual mean values).

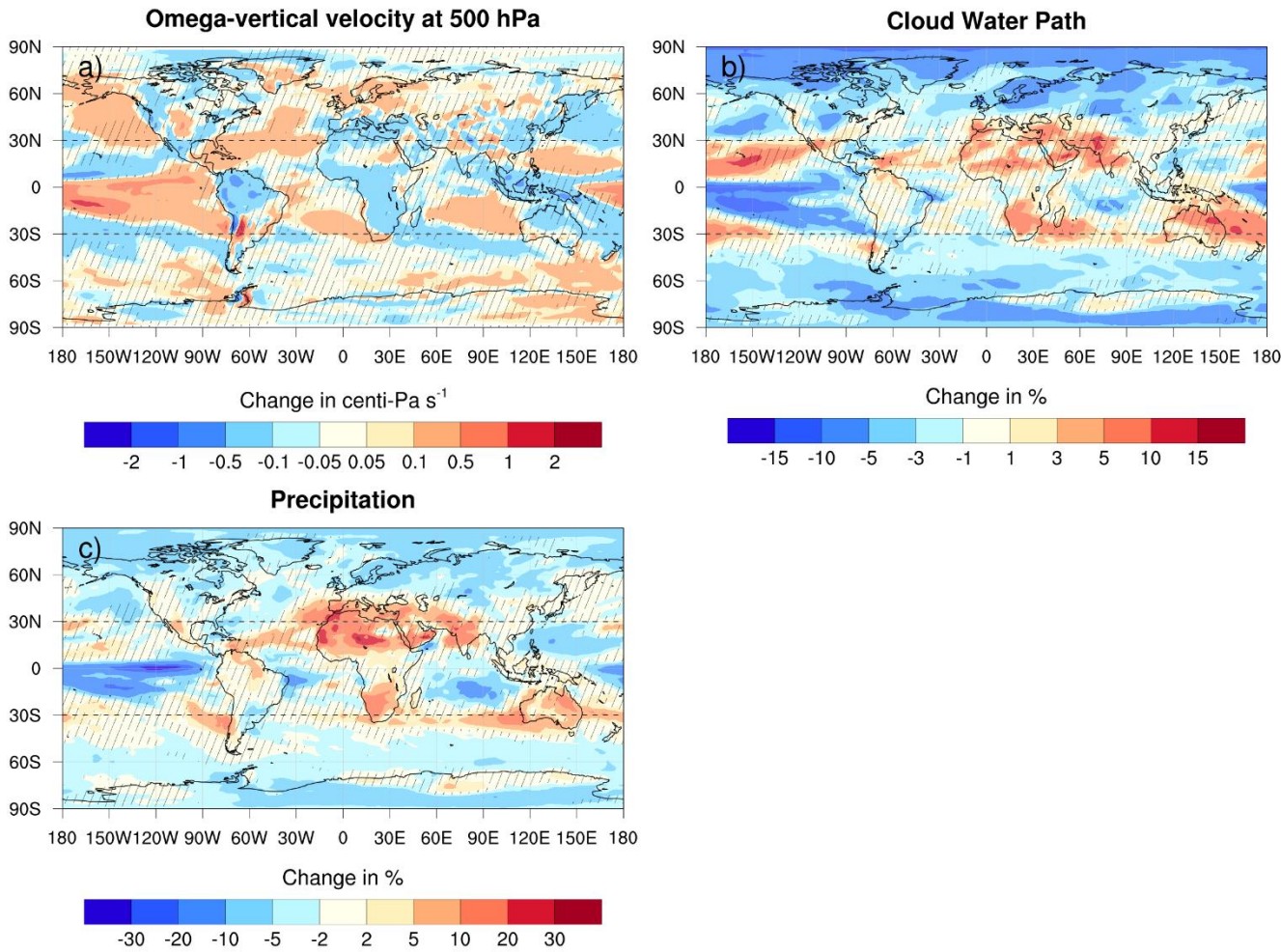

**Figure 9.** The multi-model mean difference between the G4sea-Salt experiment and RCP4.5 averaged over years 2035-2065. The multi-model mean difference refers to the mean of all three models, NorESM1-M, GISS-E2, and HadGEM2-ES. Hatching denotes areas where the models disagree on the sign of the change. Change in **a)** Omega-vertical velocity at 500 hPa (positive values corresponds to reduced upward motion) (centi-Pa s$^{-1}$), **b)** Cloud Water Path (vertically-integrated cloud water content) (%), and **c)** Precipitation rate (%).

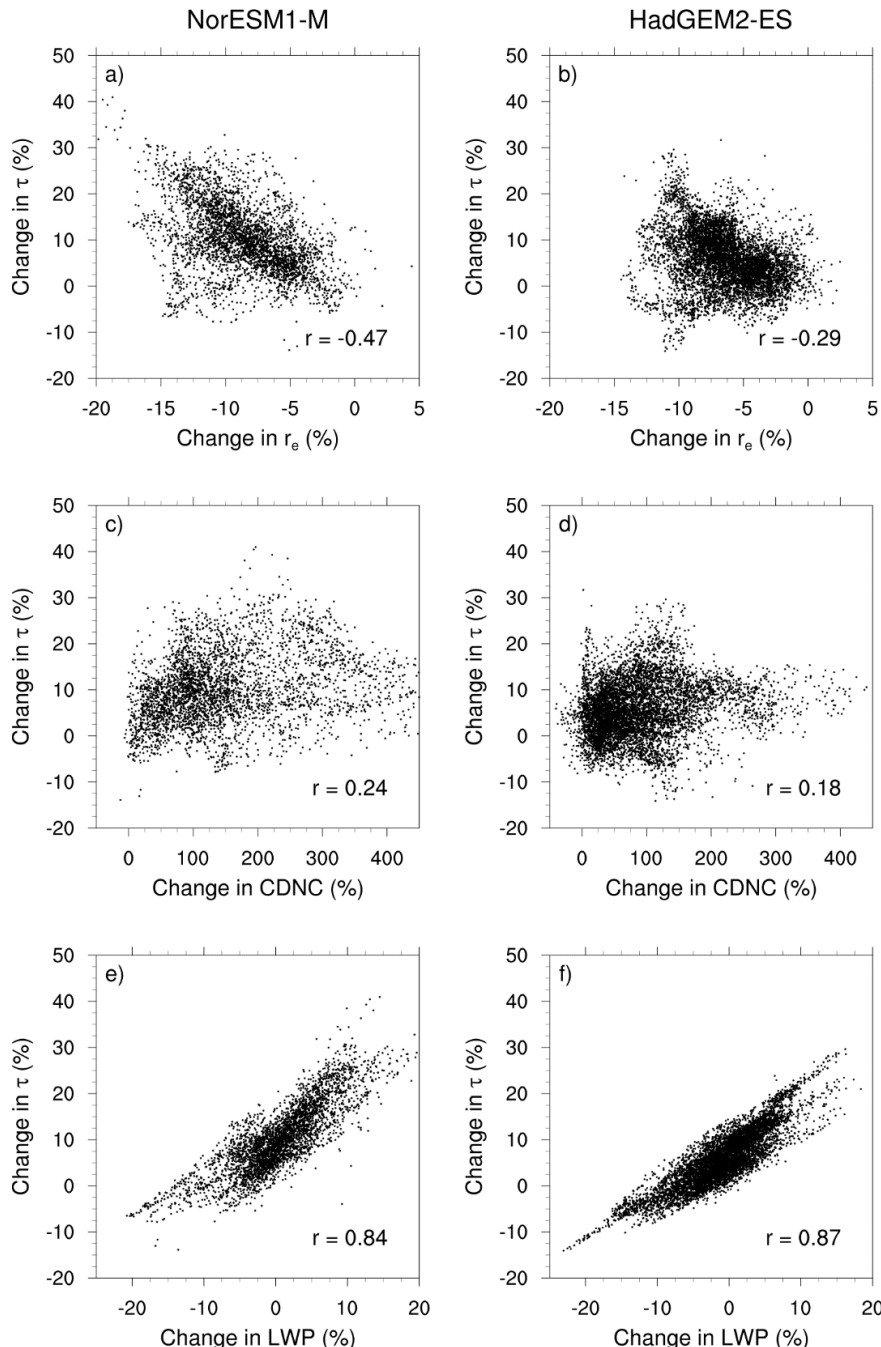

**Figure 10.** Relations between the relative change in cloud optical depth ($\tau$) due to sea salt injection against the corresponding changes in $r_e$ for **a)** NorESM1-M and **b)** HadGEM2-ES; *CDNC* for **c)** NorESM1-M and **d)** HadGEM2-ES; and *LWP* for **e)** NorESM1-M and **f)** HadGEM2-ES. The relations represent averages over the period 2035-2065 within the injection area. Pearson's correlation coefficient (*r*) is given for each relation.

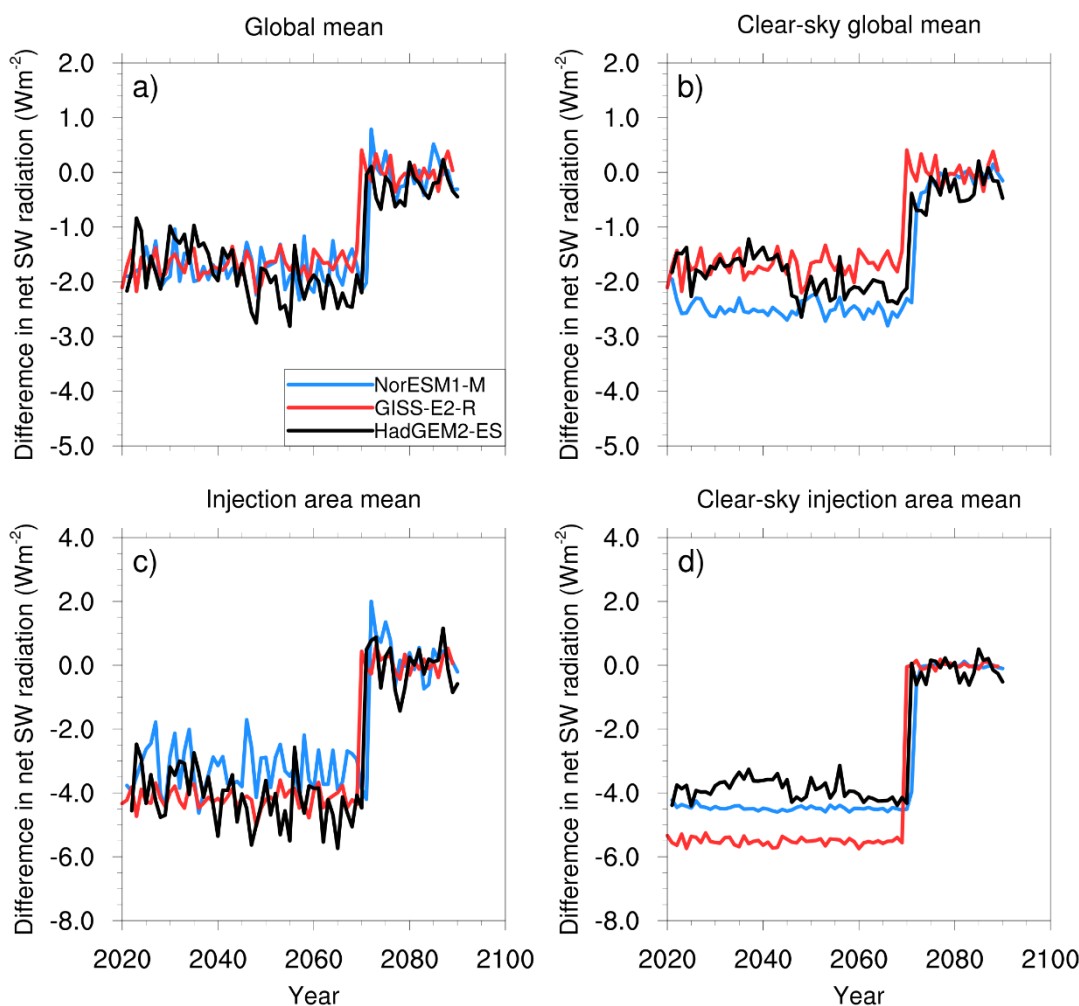

**Figure 11.** Difference in net SW radiation at the TOA between G4sea-Salt and RCP4.5 **a)** global mean, **b)** clear-sky global mean, **c)** injection area mean, and **d)** clear-sky injection area mean. The colors denote NorESM1-M (blue), GISS-E2-R (red), and HadGEM2-ES (black). Only grid cells over ocean have been included in the mean values representative of the injection area.

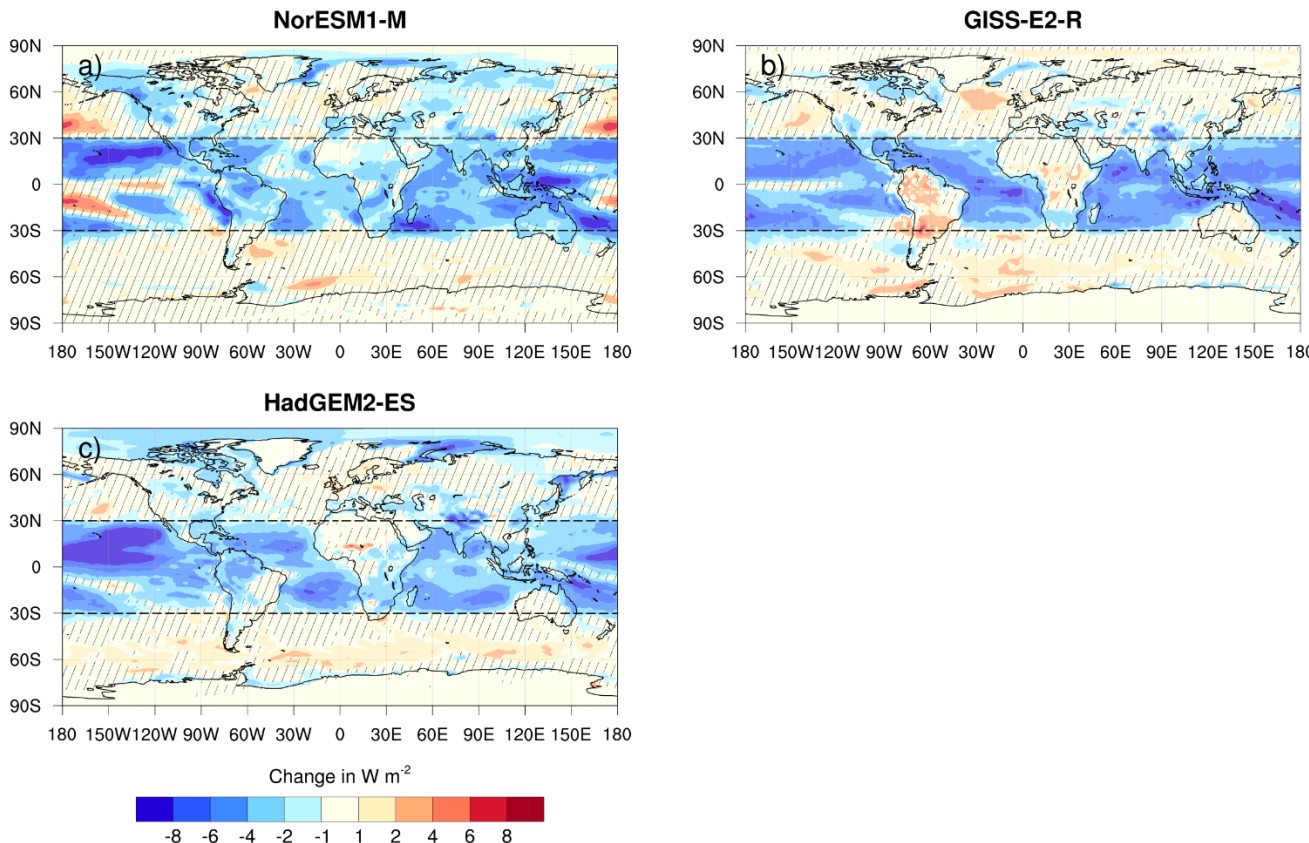

**Figure 12.** Mean change in net SW radiation [W m$^{-2}$] at the TOA between G4sea-Salt and RCP4.5 for the period 2035-2065 for **a)** NorESM1-M, **b)** GISS-E2-R, and **c)** HadGEM2-ES. Hatching denotes areas where changes are not significant at the 95% confidence level (Student t-test with respect to variance of annual mean values).