# Peer review of "Marine cloud brightening – as effective without clouds"

_Atmospheric Chemistry and Physics, 2017_

## Referee Comment (RC1) · Anonymous Referee #1 · 14 Jun 2017

Review of Marine cloud brightening – as effective without clouds, by Ahlm, Jones, Stjern, Muri, Kravitz, Egill Kristjansson

This manuscript describes a multi-model study to evaluate the efficacy of marine cloud brightening geoengineering using enhanced emissions of sea-salt particles (nominally in the accumulation mode, although the GISS model's accumulation mode has a major contribution from supermicron diameter particles). Emissions are enhanced only in the tropical belt from 30S to 30N. The simulations were conducted as part of the GeoMIP project. As with other GeoMIP comparisons, the radiative forcing from the enhanced aerosol emissions is specified, and then each model tunes its emission rate to produce a -2 W/m2 global mean effective radiative forcing (ERF). The aerosols produce both direct and indirect radiative effects, and the direct effects are found to be as large, or larger, than the indirect effects.

In my view, the direct effects produce a large fraction of the ERF for the following reasons:

1. The assumed diameter of the emitted particles is larger than that recommended in specific studies. Connolly et al. (2014), which appeared in a Phil Trans special issue on geoengineering, used detailed parcel modeling to show that median dry particle diameters from 35-100 nm are optimal for brightening marine low clouds. The diameters used in this study are 200 nm (HadGEM), 260 nm (NorESM), and 880 nm (GISS), which require at least an order of magnitude more mass to be sprayed to produce the same brightening effect (see Fig. 1b in Connolly et al. 2014). This would require enormous amounts of energy (the energy required for the production of aerosol particles scales approximately with the overall mass of salt sprayer) compared to the case with smaller particles. Small particles are most effective for brightening clouds, whereas somewhat larger particles (0.2-1 micron) are optimal for the direct effect. Given this information, one could probably have predicted that the direct forcing would be dominant before the model experiments were conducted.

2. Seeding takes place over the entire tropical ocean. Low cloud cover is limited over much of the warm tropics, so this favors direct forcing to achieve -2 W/m2 ERF. Furthermore, there is cirrus above much of the low cloud in the warm Tropics. This is not how marine cloud brightening would work in practice.

3. Cloud LWP decreases over much of the region, thus countering some of the Twomey effect.

The results presented here are quite interesting, especially the LWP responses in 3 above. However, given the choice of particle sizes used, together with a seeding strategy that would not be used in practice, leads me to question the utility of this paper. The cloud LWP reductions are interesting, but it is not clear that this pattern of response would be the same if a more appropriate seeding strategy had been used. Thus the paper is not a good strawman for how marine cloud brightening would be deployed in

practice.

No provision for separating direct from indirect effects was built into the experimental design, which is troubling, and should be addressed. APRP is one way to achieve this without re-running simulations.

Although the authors may argue that further experiments cannot be performed, I am going to recommend that the authors try their study again using particle sizes that have been shown in the literature to be appropriate for brightening clouds. I cannot recommend publication in its current form. There are a number of minor comments that I could make, but the major issues need to be addressed first.

References:

Connolly, P. J., G. B. McFiggans, R. Wood, and A. Tsiamis. "Factors Determining the Most Efficient Spray Distribution for Marine Cloud Brightening." Philosophical Transactions of the Royal Society A: Mathematical, Physical and Engineering Sciences 372, no. 2031 (November 17, 2014): 20140056–20140056. doi:10.1098/rsta.2014.0056.

---

## Author Comment (AC1) · 29 Jun 2017

We thank the reviewer for the comments on our manuscript. Please find our responses below.

Regarding the reviewer's general input on the design of the Geoengineering Model Intercomparison Project (GeoMIP) G4sea-salt experiment:

The G4sea-salt experiment was designed by Kravitz et al. (2013) to do certain things based on results that previous studies obtained. The main idea of the G4sea-salt experiment is to validate those results in a multi-model context. One of the key questions defined by Kravitz et al. (2013) is: To what extent do the effects of sea spray geoengineering depend upon the location of clouds? In this study we try to address this question. All GeoMIP simulations are idealised to a greater or lesser degree, but the point is that they form a defined, published protocol. The reviewer's stance that publication cannot be recommended because the published protocol doesn't agree with the reviewer's view seems to us miss the point of a MIP using idealised scenarios. It would be like criticizing a study of model responses in the CMIP5 instantaneous 4xCO2 experiment on the basis that such a large and rapid change in $CO_2$ is unlikely; of course this is true, but it's still an incredibly useful way of learning about climate system behaviour.

The reviewer also makes claims about how this form of geoengineering would be carried out in practice. As actual deployment is currently hypothetical, we are reluctant to make any changes to the manuscript that would alter our protocol in favour of one that is, in theory, more or less realistic.

*Specific comments to the authors*

*In my view, the direct effects produce a large fraction of the ERF for the following reasons:*

*1. The assumed diameter of the emitted particles is larger than that recommended in specific studies. Connolly et al. (2014), which appeared in a Phil Trans special issue on geoengineering, used detailed parcel modeling to show that median dry particle diameters from 35-100 nm are optimal for brightening marine low clouds. The diameters used in this study are 200 nm (HadGEM), 260 nm (NorESM), and 880 nm (GISS), which require at least an order of magnitude more mass to be sprayed to produce the same brightening effect (see Fig. 1b in Connolly et al. 2014). This would require enormous amounts of energy (the energy required for the production of aerosol particles scales approximately with the overall mass of salt sprayer) compared to the case with smaller particles. Small particles are most effective for brightening clouds, whereas somewhat larger particles (0.2-1 micron) are optimal for the direct effect. Given this information, one could probably have predicted that the direct forcing would be dominant before the model experiments were conducted.*

As mentioned above, the main goal of the G4sea-salt experiment is to validate the results that previous ESM studies have provided in a multi-model context. Existing ESM studies that take

into account the sea spray injection process include e.g. Alterskjær et al. (2012, 2013), Jones and Haywood (2012), Korhonen et al. (2010), Partanen et al. (2012), and Wang et al. (2011). In all these studies, the dry diameter of the injected particles is within the interval of 0.20-0.44 µm. Thus, the size of the injected particles in our study is within the size range of the previous ESM studies, which is necessary when validating the results of those studies.

Moreover, we did not choose this range of particle sizes simply because they were what previous studies used. Alterskjær and Kristjánsson (2013) showed using NorESM1-M that injection of accumulation mode sea salt particles resulted in a negative forcing, whereas injection of Aitken mode particles resulted in a positive forcing caused by a strong competition effect combined with high critical supersaturation of Aitken mode sea salt. Although the positive forcing caused by the injection of Aitken mode particles could be due to limitations of the Abdul-Razzak et al. scheme used in NorESM1-M, as suggested by Connolly et al. (2014), representing marine cloud brightening in our simulations requires injections that produce a negative forcing. In using such large-scale models, compromises like this must necessarily be made, but since our focus is on large-scale climate rather than process-level understanding, we are comfortable that this compromise suits the purposes of our study. Because the size distributions of the injected particles are assumed to be equal to those of the natural accumulation mode sea spray aerosol in the three models, the size of the injected particles varies across the models, allowing us to incorporate a study of model spread in our analyses.

Although the main goal of this study is to evaluate previous results, and we certainly do not claim that this study provides the exact details of how this form of geoengineering would be carried out in practice, we would still like to comment on the reviewer's statement "*The assumed diameter of the emitted particles is larger than that recommended in specific studies*". The paper by Connolly et al. (2014) is interesting, but it is only one study. In that study, it is assumed that the injected particles consist of pure sea salt. However, extensive measurements show that organics contribute substantially to the composition of sea spray aerosol, and in many areas is even the dominant constituent (e.g. de Leeuw et al., 2011). As sea spray geoengineering would likely produce particles with a similar composition as natural sea spray, the injected particles would thus need to be larger to activate to cloud droplets compared to when assuming pure sea salt as in Connolly et al. 2014. In particular, the presence of organics strongly suppresses hygroscopic growth compared to pure sea salt. This is relevant since Connolly et al. (2014) conclude that interstitial particles play an important role in controlling the albedo in their study.

Finally, the reviewer writes "*Small particles are most effective for brightening clouds, whereas somewhat larger particles (0.2-1 micron) are optimal for the direct effect. Given this information, one could probably have predicted that the direct forcing would be dominant before the model experiments were conducted*". First of all, we have not predicted that the aerosol direct effect is dominant. The conclusion of this study is that that the effective radiative forcing (ERF) by the injected particles in most regions is as large in clear-sky conditions as in cloudy-sky conditions. The exception is in the subtropical marine stratocumulus regions, in particular for HadGEM2-ES, where the presence of clouds enhances the ERF compared to clear-sky conditions. Jones and Haywood et al. (2012) obtained a much larger radiative impact when maximizing the aerosol indirect effect than when maximizing

the aerosol direct effect. The aerosol indirect effect dominated the radiative impact also in Partanen et al. (2012). Both these sea spray geoengineering studies used injections of accumulation mode particles, similar to our study. Based on previous studies, we are unable to arrive at the reviewer's conclusion.

*2. Seeding takes place over the entire tropical ocean. Low cloud cover is limited over much of the warm tropics, so this favors direct forcing to achieve -2 W/m2 ERF. Furthermore, there is cirrus above much of the low cloud in the warm Tropics. This is not how marine cloud brightening would work in practice.*

As noted above, we consider statements regarding the practical implementation of marine cloud brightening to be difficult to defend, and we are unwilling to alter our experimental protocol to conform to conjecture. The oceanic regions between 30°S and 30°N have been identified as containing most of the radiatively important stratocumulus cloud decks (Alterskjær et al., 2012; Jones and Haywood, 2012). However, we agree with the reviewer that the presence of cirrus clouds close to the equator is not optimal for sea spray geoengineering. Although high clouds likely reduce the efficiency of sea spray geoengineering, a MIP can provide information on how large this reduction is in different models, as well as how large the horizontal variability in ERF is across the injection area. The results of our study indicate that the horizontal variability may be somewhat lower than seen in, e.g., Partanen et al. (2012).

*3. Cloud LWP decreases over much of the region, thus countering some of the Twomey effect*

We agree that the LWP response is important. However, our main conclusions given in the title of the paper are based on the simulations with fixed SST. Thus, the response in LWP caused by changes in the atmospheric circulation discussed in Sect. 3.2. does not influence the ERF in Sect. 3.1.

*4. No provision for separating direct from indirect effects was built into the experimental design, which is troubling, and should be addressed. APRP is one way to achieve this without re-running simulations.*

We have been careful throughout the paper not to claim that the aerosol direct effect dominates the radiative effect, or that the aerosol direct effect would contribute as much as the aerosol indirect effect. This paper focuses (mainly) on whether the presence of clouds increases or decreases the ERF in different areas, and how much the presence of clouds influences the effects of sea spray geoengineering. It would have been troubling if we would have stated that the aerosol direct effect dominates the forcing, or if we would have tried to

estimate the contributions of the aerosol direct and indirect effects. However, that is not the focus of our study.

**References**

Alterskjær, K. et al., Atmos. Chem. Phys., 12, 2795-2807, doi:10.5194/acp-12-2795-2012, 2012.

Alterskjær, K. and Kristjánsson, J. E., Geophys. Res. Lett., 40, 210-215, doi:10.1029/2012GL054286, 2013.

Alterskjær, K. et al., J. Geophys. Res., 118, 12195-12206, doi:10.1002/2013JD020432, 2013.

Connolly, P. J., Phil. Trans. R. Soc. A, 372, 20140056, 2014.

de Leeuw, G. et al., Rev. Geophys., 49, RG2001, doi :10.1029/2010RG000349, 2011.

Jones, A. and Haywood J. M., Atmos. Chem. Phys., 12, 10887-10898, doi:10.5194/acp-12-10887-2012, 2012.

Korhonen, H. et al., Atmos. Chem. Phys., 10, 4133-4143, doi:10.5194/acp-10-4133-2010, 2010.

Kravitz et al., J. Geophys. Res., 118, 11175-11186, doi:10.1002/jgrd.50856, 2013.

Partanen, A.-I. et al., J. Geophys. Res. 117, doi:10.1029/2011JD016428, 2012.

Wang et al., Atmos. Chem. Phys., 11, 4237-4249, 2011.

---

## Referee Comment (RC2) · Anonymous Referee #2 · 3 Jul 2017

The authors compare simulations from three coupled atmosphere-ocean ESM models (NorESM1-M; GISS-E2-R; HadGEM2-ES) as part of the Geoengineering Model Inter-comparison Project to study how radiative forcing partitions between total and clear-sky effects when sea salt is injected into the atmosphere at rates sufficient to maintain a top-of-the-atmosphere (TOA) radiative forcing of -2 W/m2 relative to the RCP4.5 scenario. The simulations are carried out for the years 2020-2090, and the forced injections are maintained for the first 50 years. Important differences among the models are that the GISS model injects particles with a mean radius of 0.44 um compared with smaller particles injected by the Nor and Had models (0.13 and 0.10, respectively), and GISS simulates reduced amounts of low clouds over the injection region (30oN to 30oS). Based on the clear to total sky results, the bottom line is that these simulations suggest clear-sky forcing is comparable to the total forcing. In particular, the GISS model indicates a slightly larger clear-sky effect, the Had model showing a

slightly lower clear-sky effect and the Nor model a roughly equal effect. The authors conclude that "These findings suggest a more important role of the aerosol direct effect in sea spray climate engineering than previously thought."

The paper is well organized and well written. The result is important, but a little more insight is needed. The current results suggest another question: to what degree is optimization of particle injections necessary? Despite the factor of four difference in the mean size representation of the particle size distribution of the GISS model compared with the other two models, differences in the clear-sky forcing among the models appear to be relatively small (e.g. Fig. 2b). Considering the injection sizes, should greater differences be expected if the forcing is direct? Neither question can be considered because relatively simple explanations of fundamental particle representations used in each model are missing: 1) sub-saturated hygroscopic growth; 2) cloud activation; 3) deposition processes; 4) vertical distributions of the injected particles; 5) number size distributions of the simulated injections. The complexities and subtleties of the many aerosol processes, including effects on cloud, may offset to some degree. For example, as you know, if you try to optimize for the indirect effect by injecting particles smaller than 100 nm you expect to reduce the direct component. However, it may be difficult to either avoid spraying some larger particles or the presence of natural sea salt particles, either of which will tend to reduce the indirect effect by competition for water vapour. There is some discussion at the top of page 8, but it focusses on activation only. Some additional discussion of these processes with a focus on why the clear-sky forcing is not so different despite the substantial difference in particle representations between GISS and the other models, as well as a figure comparing injected number size distributions, would offer some insight.

Minor comments:

1) Page 2, line 31 – Should this be "an uncertainty" rather than "the uncertainty"?

2) Page 4, line 32 – Perhaps use "low-cloud amounts".

3) Page 5, lines 24-25 - How frequent are clear-sky conditions in each model?

4) Page 7, lines 11-12 and Figure 4 - Is it truly increasing or just altering the mechanism, since the ERF-TOF is held constant?

5) Page 7, line 31 - It would be more instructive to include changes in number concentrations of sea-salt particles.

6) Page 8, lines 11-14 - What are the ranges of background CDNC in each model? Why does CDNC over northern Greenland reduce so much in NorESM, and over the high Arctic in HadGEM2?

7) Page 9, lines 8-10 - Of course the relative impact of LWP is well known. What would be helpful is to know how Figures 9c and 9d compare with observations, if there are sufficient data to do that.

8) A note - Sea salt particles of 0.88 um diameter (GISS) or larger will be very hard to activate (by definition) in clouds. To reach their activation point they need to take up a very large amount of water, and that may not happen.
* * *

---

## Author Comment (AC2) · 30 Aug 2017

We thank the reviewer for the constructive comments and suggestions for improvement of our manuscript. The comments from the reviewer followed by our responses to the comments can be seen below.

*Major comment*

*The paper is well organized and well written. The result is important, but a little more insight is needed. The current results suggest another question: to what degree is optimization of particle injections necessary? Despite the factor of four difference in the mean size representation of the particle size distribution of the GISS model compared with the other two models, differences in the clear-sky forcing among the models appear to be relatively small (e.g. Fig. 2b). Considering the injection sizes, should greater differences be expected if the forcing is direct? Neither question can be considered because relatively simple explanations of fundamental particle representations used in each model are missing: 1) sub-saturated hygroscopic growth; 2) cloud activation; 3) deposition processes; 4) vertical distributions of the injected particles; 5) number size distributions of the simulated injections. The complexities and subtleties of the many aerosol processes, including effects on cloud, may offset to some degree. For example, as you know, if you try to optimize for the indirect effect by injecting particles smaller than 100 nm you expect to reduce the direct component. However, it may be difficult to either avoid spraying some larger particles or the presence of natural sea salt particles, either of which will tend to reduce the indirect effect by competition for water vapour. There is some discussion at the top of page 8, but it focuses on activation only. Some additional discussion of these processes with a focus on why the clear-sky forcing is not so different despite the substantial difference in particle representations between GISS and the other models, as well as a figure comparing injected number size distributions, would offer some insight.*

Response:

We agree with the reviewer that the size of the injected particles is relevant for the clear-sky effective radiative forcing (ERF). The injected sea salt particles within the G4sea-salt experiment have a median dry radius of 0.13 μm in NorESM1-M, 0.44 μm in GISS-E2-R, and 0.10 μm in HadGEM2-ES. However, the fact that the clear-sky global-mean ERF in Fig. 2b in the manuscript (Fig. 3b in the updated version) is similar in magnitude for the three models does not imply that the difference in particle size between the models has a negligible effect on the clear-sky ERF. The reason that such a conclusion cannot be drawn is that the sea salt injection rates are not equal for the three models, since the injection rates have been set to generate a total global-mean ERF of -2.0 W m$^{-2}$ in each model. The injection rates are 250 Tg yr$^{-1}$ in NorESM1-M, 590 Tg yr$^{-1}$ in GISS-E2-R, and 200 Tg yr$^{-1}$ in HadGEM2-ES, as mentioned in Sect. 3.2. Thus, the injection rates vary with almost a factor of three across the three models.

The size distributions of the sea salt injection are shown in Figure 1 for particle number (Fig. 1a), particle surface area (Fig. 1b), and particle mass (Fig. 1c). This figure is now

included in the updated manuscript as suggested by the reviewer. The mass scattering efficiency of homogeneous spheres of sea salt (refractive index = 1.544) for light with a wavelength of 550 nm peaks for a particle radius of ~0.3 μm (Seinfeld and Pandis, 1998). Thus, for a constant mass concentration, maximum scattering efficiency is expected for particle sizes somewhere in between the median dry radius of the injections in GISS-E2-R and the median dry radii of the injections in NorESM1-M and HadGEM2-ES. However, as mentioned above (and shown in Fig. 1c below), mass concentrations are not equal in the models since the injected sea salt mass is larger in GISS-E2-R than in the other two models. On top of this, aerosol processing in the atmosphere, transport, and deposition will affect the clear-sky forcing, as pointed out by the reviewer. We have added some more description of these processes into Sect. 2.1 of the updated version of the manuscript.

[Figure]

**Figure 1.** Size distributions for the total sea salt injections (30°N and 30°S) of a) particle number $I_N$, b) particle surface area $I_S$, and c) particle mass $I_M$ for NorESM1-M (blue), GISS-E2-R (red), and HadGEM2-ES (green).

In Fig. 1 above, the particle surface area (Fig. 1b) is the variable that is closest related to the amount of light scattered by the sea salt particles (and thereby the clear-sky ERF in Fig.

2b in the manuscript, or Fig. 3b in the updated version). For a full description of Mie scattering, however, one needs to take into account also variations in the scattering coefficient with particle size, which is done in the radiative transfer calculations in the models. The total particle number injections (integrated over the number size distributions in Fig. 1a) are $1.8 \cdot 10^{20}$, $2.7 \cdot 10^{18}$, and $1.1 \cdot 10^{20}$ s$^{-1}$ for NorESM1-M, GISS-E2-R, and HadGEM2-ES, thus almost two orders of magnitude smaller number injection in GISS-E2-R compared to the other models. The corresponding particle surface injections (integrated over the particle surface distributions in Fig. 1b) are $5.2 \cdot 10^7$, $1.7 \cdot 10^7$, and $3.1 \cdot 10^7$ m$^2$ s$^{-1}$ for NorESM1-M, GISS-E2-R, and HadGEM2-ES. Thus, although the difference in total particle number injection between GISS-E2-R and the other two models is large, the difference in total particle surface area injection is considerably smaller. Based on these numbers it is not so surprising that the clear-sky ERF is rather equal in magnitude for the three models.

[Figure]

**Figure 2.** Mean clear-sky optical depth of the atmosphere for RCP4.5 between 2035 and 2065 for a) NorESM1-M, b) GISS-E2-R, and c) HadGEM2-ES.

When it comes to the processes listed by the reviewer, dry deposition of the injected particles should be slightly faster in GISS-E2-R than in the other two models, since the injected particles are larger in GISS-E2-R, which implies somewhat higher dry deposition velocities due to more efficient interception and impaction, and even gravitational settling for the largest particles. Vertical mixing is more efficient in NorESM1-M than in the other models, but this is likely of minor importance for the clear-sky ERF, although it may be

relevant for the aerosol direct effect if a fraction of the injected sea salt particles can be transported above the stratocumulus layers.

However, the main reason that the somewhat smaller particle surface injections in GISS-E2-R still generates a clear-sky ERF as large as the other two models (or even slightly larger) is likely due to GISS-E2-R having the lowest background clear-sky atmospheric optical depth of the three models (Fig. 2). This means that GISS-E2-R is more sensitive to injections than the two other models. The lower clear-sky atmospheric optical depth in GISS-E2-R is at least to some extent related to lower background sea salt concentrations. The background sea salt mass concentrations within the injection area for RCP4.5 (2035-2065) at the lowest model layer are 14.4, 7.9, and 54.1 μg m$^{-3}$ for NorESM1-M, GISS-E2-R, and HadGEM2-ES, respectively. The impact of mineral dust outflow from Africa over the Atlantic Ocean on the clear-sky optical depth (Fig. 2) is also less pronounced in GISS-E2-R than in the other two models, which should also contribute to a higher sensitivity to sea salt injections in GISS-E2-R. Some of this discussion has been added to the updated version of the manuscript.

*Minor comments*

*Comment #1*

*Page 2, line 31 – Should this be "an uncertainty" rather than "the uncertainty"?*

Response:

Thanks, we have changed this.

*Comment #2*

*Page 4, line 32 – Perhaps use "low-cloud amounts".*

Response:

Thanks, changed.

*Comment #3*

*Page 5, lines 24-25 - How frequent are clear-sky conditions in each model?*

Response:

This is a relevant question because if there were large regions frequently dominated by clear-sky, it would not be surprising that the ERF by the injected particles in these areas are similar

in total and in clear-sky conditions. However, from Fig. 1 in the paper (Fig. 2 in the updated version) we know that almost everywhere within the injection area, the mean cloud fraction of low-level clouds is larger than 40% and 30% in NorESM1-M and HadGEM2-ES, respectively. As mentioned in the manuscript, the low-level cloud fraction is considerably smaller in GISS-E2-R than in the other two models.

As we only have cloud cover model output as monthly mean values, it is not possible to tell how frequent clear sky conditions are. During a period of a month, there will be days with clouds in all locations which implies that the monthly mean cloud fraction is never zero in any marine location. However, Fig. 2 below shows the percentage fraction of all months between 2035 and 2065 with a total cloud fraction less than 50% within the injection area for RCP4.5 for the three models. As seen in the figure, months with a mean total cloud fraction below 50% are most frequent in HadGEM2-ES. Note that the total cloud fraction includes cloud layers at all heights. If only low-level clouds are included, GISS-E2-R has the highest frequency of low cloudiness, as discussed in Sect. 3 in the paper.

[Figure]

**Figure 2.** Percentage of all months between 2035 and 2065 with a mean total cloud cover below 50% in the injection area for NorESM1-M, GISS-E2-R, and HadGEM2-ES.

*Comment # 4*

*Page 7, lines 11-12 and Figure 4 - Is it truly increasing or just altering the mechanism, since the ERF-TOF is held constant?*

Response:

We do not fully understand this comment by the reviewer. What is held constant is the sea salt injection rates. These constant injection rates generate an ERF that is more or less constant with time. The global-mean clear-sky ERF is not held constant at a certain value, but happens to be almost equal in magnitude to the global-mean cloudy-sky ERF. However, the fact that the global-mean total ERF is almost equal to the corresponding clear-sky ERF does not imply that these are equal in all locations. In the subtropical high pressure cells, the presence of low-level clouds increases the regional ERF compared to clear-sky conditions, in particular in HadGEM2-ES. In contrast, closer to the equator the presence of high-level clouds decreases the ERF compared to clear-sky conditions, in particular in GISS-E2-R.

*Comment #5*

*Page 7, line 31 - It would be more instructive to include changes in number concentrations of sea-salt particles.*

Response:

We agree with the reviewer. Unfortunately, the particle number concentration is only diagnosed in NorESM1-M, which is the reason why this variable is not shown in the paper. The change in number concentration due to sea spray climate engineering in NorESM1-M is shown in Fig. 3 below. As can be seen in this figure, the sentence that the reviewer refers to in the manuscript ("In NorESM1-M (Fig. 5a), comparatively large increases in sea salt concentration occur in the subtropical high pressure regions") is valid also for the particle number concentration.

*Comment #6*

*Page 8, lines 11-14 - What are the ranges of background CDNC in each model? Why does CDNC over northern Greenland reduce so much in NorESM, and over the high Arctic in HadGEM2?*

Response:

The background CDNC within the injection area at an altitude of ~1000 m averaged over 2035-2065 for RCP4.5 varies for NorESM1-M from 10-20 cm$^{-3}$ in the remote areas of Pacific

and reaches a maximum of ~100 cm$^{-3}$ south of Mexico, west of Northern Africa, south-east of China, and over the northern parts of the Indian Ocean. HadGEM2-ES has its maxima in CDNC at similar locations within the injection area. However, HadGEM2-ES has somewhat higher concentrations with a typical CDNC of 20-40 cm$^{-3}$ in the remote Pacific Ocean and CDNC reaching 250 cm$^{-3}$ at coastal locations closer to continental sources. GISS-E2-R has higher background CDNC than the other models with concentrations of 50-100 cm$^{-3}$ in the remote Pacific Ocean and concentrations higher than 1000 cm$^{-3}$ in some coastal regions influenced by continental sources. Whereas NorESM1-M and HadGEM2-ES predict CDNC close to estimates using MODIS data for cloud top CDNC (e.g. Wood, 2012), GISS-E2-R predicts higher background CDNC than estimated from MODIS. The relatively high background CDNC in GISS-E2-R is the reason for the smaller percentage increase in CDNC due to sea spray climate engineering in GISS-E2-R compared to the other two models.

Concerning the reduction in CDNC over the Arctic region in NorESM1-M and HadGEM2-ES, the variable CDNC represents the number concentration of cloud liquid water particles in the air, and the CDNC is lower than 1 cm$^{-3}$ over Greenland in NorESM1-M and just slightly higher than 1 cm$^{-3}$ in HadGEM2-ES. Thus, a very small absolute change in concentration can result in a very large relative change in CDNC. The explanation for the reduction in CDNC is probably that the G4sea-salt experiment results in a cooling of the Arctic region, which implies less liquid water in the clouds over e.g. Greenland. Another mechanism for the reduction of CDNC in the Arctic is also related to the cooling induced by the sea-salt: the cooling increases the sea-ice cover in the Arctic and therefore reduces the source of natural sea salt and Dimethyl sulphide (DMS), both of which cause a reduction in CDNC. We have added this information to the manuscript.

[Figure]

**Figure 3.** Difference in particle number concentration between G4sea-salt and RCP4.5 in NorESM1-M averaged over the period 2035-2065.

*Comment #7*

*Page 9, lines 8-10 - Of course the relative impact of LWP is well known. What would be helpful is to know how Figures 9c and 9d compare with observations, if there are sufficient data to do that.*

Response:

The point we want to make with Fig. 9 is that for these long time scales (sea spray climate engineering for 15-45 years), changes in the atmospheric circulation and the resulting changes in LWP will be the main controller of the cloud optical depth in a certain location, rather than changes in CDNC. Therefore, this is not something that can be compared to observations.

*Comment #8*

*A note - Sea salt particles of 0.88 um diameter (GISS) or larger will be very hard to activate (by definition) in clouds. To reach their activation point they need to take up a very large amount of water, and that may not happen.*

Response:

We agree with the reviewer that the aerosol indirect effect would likely be favoured by injections of particles with a smaller size than those injected in GISS-E2-R. However, a benefit of applying somewhat varying sizes for the sea salt injections in the different models is that it allows us to incorporate a study of model spread in our analyses.

**References**

Wood, R.: Stratocumulus clouds, Monthly Weather Review, 140, 2373-2423, 2012.

---

## Author Response (AR2)

**Author reply to the comment by the referee**

Dear Editor,

Thanks for the report. Regarding point 1 by the reviewer, we have added the following text to Sect. 3.1 of the manuscript:

[revised manuscript text omitted]